# Composite regulation of ERK activity dynamics underlying tumour-specific traits in the intestine

Yu Muta[1,2], Yoshihisa Fujita[3], Kenta Sumiyama[4], Atsuro Sakurai[5], M. Mark Taketo[6], Tsutomu Chiba[2,7], Hiroshi Seno[2], Kazuhiro Aoki[8,9], Michiyuki Matsuda[1,5] & Masamichi Imajo[5]

Acting downstream of many growth factors, extracellular signal-regulated kinase (ERK) plays a pivotal role in regulating cell proliferation and tumorigenesis, where its spatiotemporal dynamics, as well as its strength, determine cellular responses. Here, we uncover the ERK activity dynamics in intestinal epithelial cells (IECs) and their association with tumour characteristics. Intravital imaging identifies two distinct modes of ERK activity, sustained and pulse-like activity, in IECs. The sustained and pulse-like activities depend on ErbB2 and EGFR, respectively. Notably, activation of Wnt signalling, the earliest event in intestinal tumor-igenesis, augments EGFR signalling and increases the frequency of ERK activity pulses through controlling the expression of EGFR and its regulators, rendering IECs sensitive to EGFR inhibition. Furthermore, the increased pulse frequency is correlated with increased cell proliferation. Thus, ERK activity dynamics are defined by composite inputs from EGFR and ErbB2 signalling in IECs and their alterations might underlie tumour-specific sensitivity to pharmacological EGFR inhibition.

[1] Department of Pathology and Biology of Diseases, Graduate School of Medicine, Kyoto University, Kyoto 606-8051, Japan. [2] Department of Gastroenterology and Hepatology, Graduate School of Medicine, Kyoto University, Kyoto 606-8507, Japan. [3] Department of Systems Science, Graduate School of Informatics, Kyoto University, Kyoto 606-8501, Japan. [4] Laboratory for Mouse Genetic Engineering, Quantitative Biology Center, RIKEN, Osaka 565-0874, Japan. [5] Laboratory of Bioimaging and Cell Signaling, Graduate School of Biostudies, Kyoto University, Kyoto 606-8501, Japan. [6] Division of Experimental Therapeutics, Graduate School of Medicine, Kyoto University, Kyoto 606-8501, Japan. [7] Kansai Electric Power Hospital, Osaka 553-0003, Japan. [8] Division of Quantitative Biology, Okazaki Institute for Integrative Bioscience, National Institute for Basic Biology, National Institutes of Natural Sciences, Okazaki, Aichi 444-8787, Japan. [9] Department of Basic Biology, Faculty of Life Science, SOKENDAI (Graduate University for Advanced Studies), Myodaiji, Okazaki, Aichi 444-8787, Japan. Correspondence and requests for materials should be addressed to M.I. (email: mimajo@lif.kyoto-u.ac.jp)

The extracellular signal-regulated kinase (ERK) signalling pathway regulates a variety of biological processes including cell proliferation, survival, differentiation, and tumorigenesis[1, 2]. Since ERK activation promotes proliferation of many types of cells, its deregulated/constitutive activation is often observed in various cancers. Among many growth factor receptors, epidermal growth factor receptor (EGFR) plays a pivotal role in activating ERK in normal and cancerous epithelia[3], therefore, EGFR–ERK signalling has been of particular interest in cancer biology[4, 5]. In the classical view, EGF stimulation simply triggers transient and short-lived ERK activation[1, 6]. However, recent studies using a highly sensitive biosensor for ERK activity[7] have revealed that EGF signalling can generate complex spatio-temporal ERK activity at the single cell level[8–10]. For instance, certain types of cultured cells show considerable heterogeneity in ERK activity due to spontaneous ERK activation pulses and its lateral propagation to adjacent cells, both of which were associated with cell proliferation[8, 10]. Similarly, propagation of ERK activity and its correlation with cell proliferation were also observed in the mouse skin[11]. Notably ERK activity dynamics as well as its overall strength can be a critical determinant of cell proliferation[8, 9]. Moreover, difference in ERK activity dynamics leads to different outputs in some biological processes. For example, in PC12 cells, treatment with NGF or FGF induces prolonged ERK activation and neuronal differentiation[12, 13], whereas EGF treatment generates only transient, pulse-like ERK activation without inducing the differentiation[13]. Despite its obvious importance, however, how ERK activity dynamics are regulated and how they affect the physiological processes remains unknown.

The intestinal epithelium is one of the representative tissues in which EGFR–ERK signalling regulates both normal homoeostasis and tumorigenesis[14]. In this tissue, actively dividing stem cells expressing a marker gene, LGR5, support the rapid and constant renewal of the entire epithelium[15]. In mice, depletion of either EGFR or three of its ligands, EGF, amphiregulin, and TGF-α, impairs proliferation of intestinal stem cells (ISCs) and progenitor cells[16, 17]. Conversely, excessive activation of EGFR signalling by depleting its negative regulator, Lrig1, induces ISC expansion and tumour development[18, 19]. In line with these findings, EGFR signalling has also been implicated in human colorectal cancer (CRC). Typically, sporadic CRCs develop through the adenoma–carcinoma sequence, an archetypal model of multi-step carcinogenesis[20]. In this model, mutations in the adenomatous polyposis coli (APC) gene have been regarded as the earliest and the rate-limiting events of tumour initiation. Following APC mutations, sequential accumulation of other genetic mutations including KRAS, BRAF, PIK3CA, SMAD4, and TP53 mutations transforms the tissue to malignant tumours[20–22]. In addition, EGFR overexpression is also observed in human CRCs, and is associated with poor prognosis[23–26]. Pharmacological inhibition of EGFR signalling has been shown to be effective against these cancers[27]. However, mutations in KRAS or BRAF desensitize CRCs to EGFR inhibition[28], suggesting that RAS-RAF-ERK signalling mediates the tumour-promoting activity of EGFR signalling. Collectively, these reports suggest that EGFR–ERK signalling is a key driver of stem/progenitor cell proliferation and tumour progression in the intestinal epithelium in both mice and humans. However, EGFR–ERK signalling dynamics and their regulatory mechanisms remain unknown due to technical difficulties.

Recent advances in detecting ERK activity using fluorescent biosensors and culturing primary intestinal epithelial cells (IECs) as organoids[29] have paved the way to visualize EGFR–ERK signalling dynamics in this tissue. Since intestinal organoids comprise IECs without any genetic mutations and can be cultured in serum-free media, dynamic regulation of the EGFR–ERK pathway and its interaction with other pathways can be readily analyzed. Here, by taking the full advantage of the organoid culture method and a highly sensitive biosensor for ERK activity, we uncover the ERK activity dynamics in IECs. We demonstrate the presence of two distinct modes of ERK activity, sustained, constant activity and pulse-like activity, both in vivo and in vitro. Our analyses show that the two modes of ERK activity are generated by different EGFR family receptors. Moreover, we reveal that Wnt signalling activation alters the ERK signalling dynamics, which underlies the enhanced responsiveness of tumour cells to EGFR inhibition.

## Results

**In vivo imaging of ERK activity in the mouse small intestine.** To reveal the ERK activity dynamics in the intestinal epithelium, we used transgenic mice ubiquitously expressing a highly sensitive Förster resonance energy transfer (FRET) biosensor for ERK activity (EKAREV-NLS) (Fig. 1a)[30]. The small intestine of EKAREV-NLS mice was observed under an inverted two-photon excitation microscope (Fig. 1b). By this approach, ERK activity represented by the FRET/CFP ratio could be live-imaged at a single-cell resolution in areas ranging from the crypt bottom to the villus (Supplementary Fig. 1a). To validate the specificity of the biosensor, we intravenously administered a known activator of the ERK pathway, 12-O-tetradecanoylphorbol-13-acetate (TPA), or a MEK inhibitor, PD0325901. As expected, TPA increased the FRET/CFP ratio, whereas the MEK inhibitor decreased it (Fig. 1c–f), indicating that EKAREV-NLS faithfully monitors ERK activity. Thus, hereafter, we use the FRET/CFP ratio as an index of ERK activity. In vivo time-lapse imaging then revealed that IECs at the crypt exhibited sporadic ERK activity pulses (Fig. 1g–j and Supplementary Movie 1). The ERK activity pulses fired spontaneously in each cell (Fig. 1g, h and Supplementary Fig. 1b) or in some cases were propagated from adjacent cells within single crypt units (Fig. 1i, j and Supplementary Fig. 1c, d). The duration of single ERK activity pulses and the velocity of propagation in IECs were comparable to those in cultured cells[8]. These results demonstrate two modes of ERK activity in the intestinal epithelium: the sustained, basal activity that was evident by the decrease in ERK activity after MEK inhibitor treatment and the pulse-like activity that may arise spontaneously in each cell or be propagated from adjacent cells.

**ERK activity dynamics in intestinal organoids.** To facilitate the analysis of ERK activity dynamics in IECs, we generated intestinal organoids from EKAREV-NLS mice and cultured them in medium containing EGF, Noggin, and R-spondin 1 (ENR medium) (Fig. 2a, b). The growth rate and morphology of the EKAREV-NLS organoids were indistinguishable from those of wild-type organoids. As observed in vivo, TPA increased the FRET/CFP ratio (Fig. 2c–e), whereas PD0325901, a potent MEK inhibitor, decreased it (Fig. 2f–h). The FRET/CFP ratios correlated well with the fraction of phosphorylated ERK (Supplementary Fig. 2a). In addition, we also observed the expected effects of a BRAF inhibitor (BRAFi) on basal ERK activity (Fig. 2i): high concentration of BRAFi decreased the basal ERK activity, whereas low concentration of BRAFi contrastingly increased it, a phenomenon known as paradoxical activation[31]. Collectively, these results demonstrate that the EKAREV-NLS biosensor can monitor ERK activity in intestinal organoids as sensitively as in vivo.

We next examined whether the spatiotemporal dynamics of ERK activity in vivo are recapitulated in intestinal organoids. Time-lapse imaging of intestinal organoids revealed that most cells exhibited ERK activity pulses as observed in vivo (Fig. 3a–c,

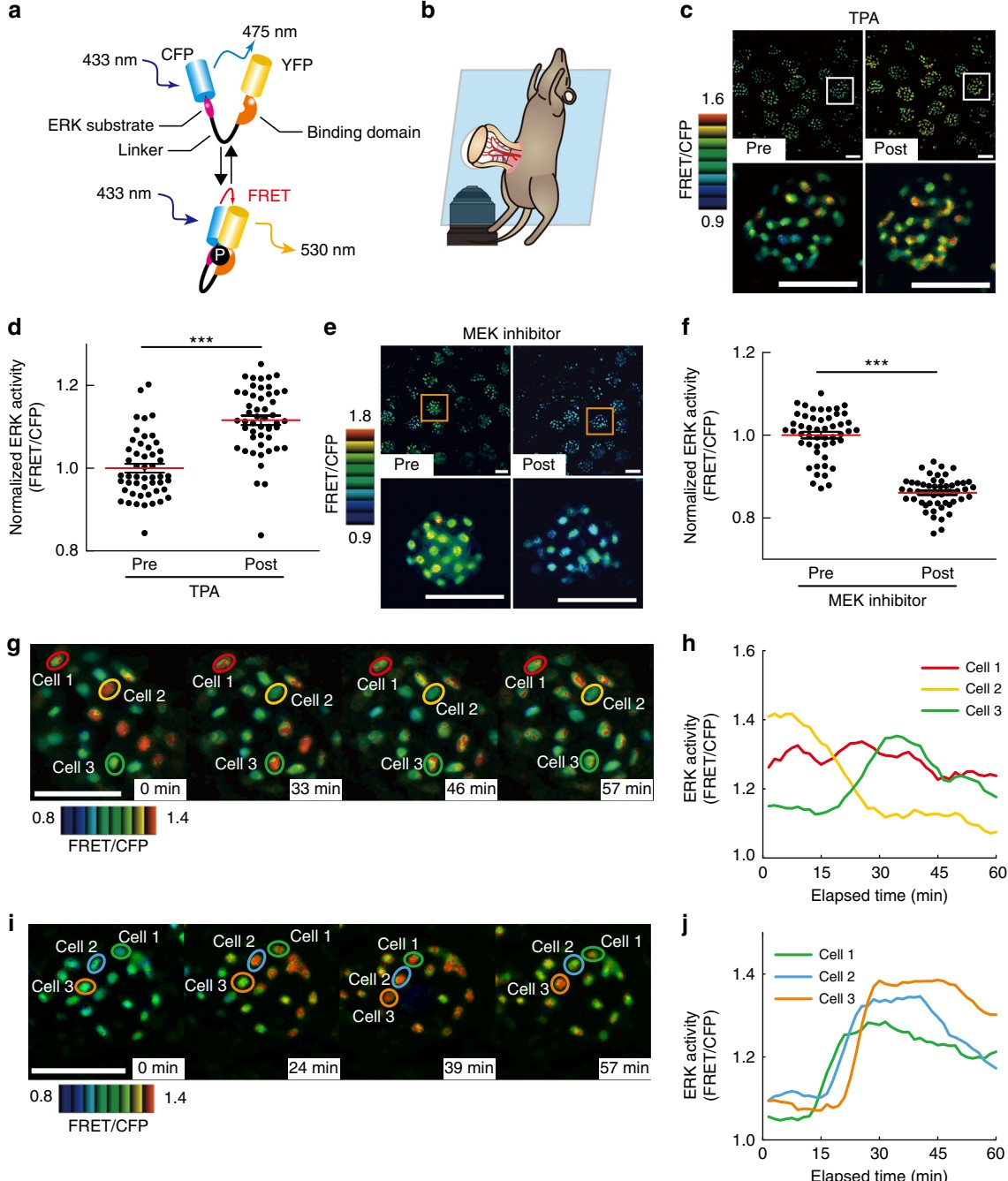

**Fig. 1** In vivo imaging of ERK activity dynamics in the mouse small intestine. **a** Schematic representation of the structure of the FRET biosensor for ERK activity, EKAREV-NLS, and its mechanism of action. Upon activation, ERK phosphorylates the substrate sequence in the biosensor. The WW domain specifically binds to the phosphorylated substrate, which brings CFP and YFP into close proximity. In this "closed" conformation state, excitation energy absorbed by CFP is transferred to YFP without radiation, thereby enabling YFP to emit fluorescence. The biosensor returns from the "closed" to the original "open" conformation by phosphatase-dependent dephosphorylation of the substrate sequence. **b** Experimental setting of in vivo imaging of the small intestine. The mouse small intestine was exteriorized, fixed on the microscope stage, and observed with an inverted two-photon excitation microscope under inhalation anaesthesia. **c** The representative FRET/CFP images of the small intestine of EKAREV-NLS mice before and after administration of 0.1 mg kg$^{-1}$ body weight of TPA. **d** Bee swarm plots showing the ERK activity (FRET/CFP ratio) in each cell before and after the TPA ($n = 50$ cells pooled from three crypts). **e** The representative FRET/CFP images of the small intestine of EKAREV-NLS mice before and after administration of 5 mg kg$^{-1}$ body weight of a MEK inhibitor (PD0325902). **f** Bee swarm plots showing the FRET/CFP ratios in each cell before and after MEK inhibitor treatment ($n = 50$ cells pooled from three crypts). **g–j** In vivo time-lapse imaging of intestinal crypts of EKAREV-NLS mice. The representative images (**g**, **i**) and quantified data from three selected cells (cells 1–3) (**h**, **j**) are shown. Note that the ERK activity pulses are spontaneously generated (**g**, **h**), or propagated from adjacent cells (**i**, **j**). The original time course data of ERK activity used for **h** and **j** before smoothing by the moving average are shown in Supplementary Fig. 1b, c. Scale bars, 50 μm. Red lines represent mean. Error bars represent s.e.m. Mann–Whitney $U$-tests were used for comparison (**d**, **f**). *$P < 0.05$, **$P < 0.001$, ***$P < 0.0001$

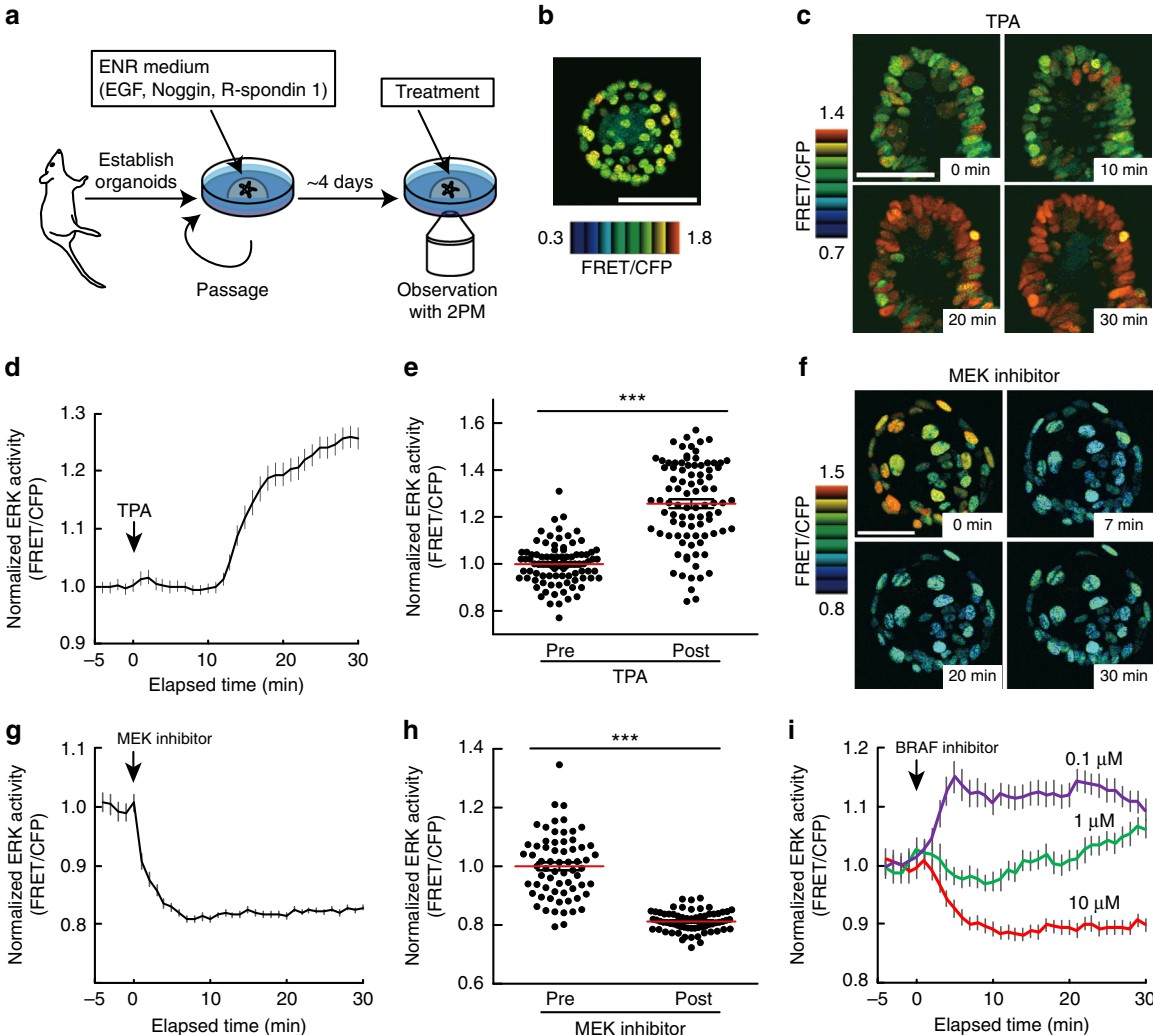

**Fig. 2** Characterization of mouse intestinal organoids expressing the FRET biosensor for ERK activity. **a** Experimental setting for live imaging of intestinal organoids. Organoids were established from the small intestine of EKAREV-NLS mice, and cultured in media containing EGF, Noggin, and R-spondin1. Imaging was performed within 4 days of the last passage. **b** A representative FRET/CFP ratio image of an EKAREV-NLS organoid cultured under normal conditions. **c**–**h** Time-lapse imaging of ERK activity in the EKAREV-NLS organoids. Organoids were treated with 1 μM TPA (**c**–**e**) or 200 nM MEK inhibitor (PD0325901) (**f**–**h**) at time point 0. **c**, **f** Representative time-lapse images of organoids treated with TPA (**c**) or a MEK inhibitor (**f**) are shown. **d**, **g** Time courses of the average FRET/CFP values in the organoids (n = 90 (**d**) and 55 (**g**) cells). The FRET/CFP values in individual cells were normalized to the mean values before the treatment. **e**, **h** Bee swarm plots of the FRET/CFP values in each cell before and after the treatment (n = 90 (**e**) and 66 (**h**) cells). The FRET/CFP values in individual cells were normalized to the mean values before the treatment. **i** Time courses of the average ERK activity (FRET/CFP ratio) in EKAREV-NLS organoids treated with 10 μM, 1 μM, or 100 nM of a BRAF inhibitor (SB590885) (n = 30, 35, and 31 cells, respectively). Scale bars, 50 μm. Red lines represent mean. Error bars represent s.e.m. Mann–Whitney *U*-tests were used for comparison (**e**, **h**). *$P < 0.05$, **$P < 0.001$, ***$P < 0.0001$

Supplementary Fig. 2b, c, and Supplementary Movie 2). Temporal autocorrelation analysis of ERK activity did not show any apparent periodicity (Fig. 3d), suggesting the stochastic nature of the pulses. Although ERK activity pulses occurred spontaneously in individual cells, propagation of the pulses was not observed in intestinal organoids cultured in the ENR medium, which contains high concentrations of EGF. Since propagation of ERK activity was mediated by shedding of EGFR ligands in cultured cells[8, 32, 33], we reasoned that EGF supplemented in the ENR medium decreased the sensitivity of cells to the lower amount of EGFR ligands secreted by cells through negative-feedback mechanisms[34–36]. To test this idea, the organoids were cultured in the absence of EGF (in the NR medium). No difference was observed in the basal ERK activity (Fig. 3e), probably due to adaptation. As expected, organoids cultured in NR, but not in ENR, exhibited ERK activation upon EGF addition (Fig. 3f). Furthermore, organoids cultured in NR occasionally exhibited cell-to-cell

propagation of ERK activity pulses (Fig. 3g, h, Supplementary Fig. 2d, and Supplementary Movie 3). Propagation of ERK activity pulses was diminished either by an EGFR inhibitor, PD153035, or by a broad-spectrum matrix metalloproteinase inhibitor, marimastat, which should suppress shedding of EGFR ligands[37, 38] (Fig. 3i, j). These results suggest that, in intestinal organoids, propagation of ERK activity pulses requires shedding of EGFR ligands and the resulting EGFR activation. Thus, the ERK activity dynamics observed in the intestinal epithelium were successfully recapitulated in intestinal organoids.

**ErbB2 and EGFR generate distinct modes of ERK activity**. We next investigated the upstream signalling pathways that contribute to ERK activity dynamics. Because EGFR and ErbB2 (a.k.a human EGFR2, HER2) are often overexpressed in gastrointestinal cancers and agents targeting these receptors have been used in

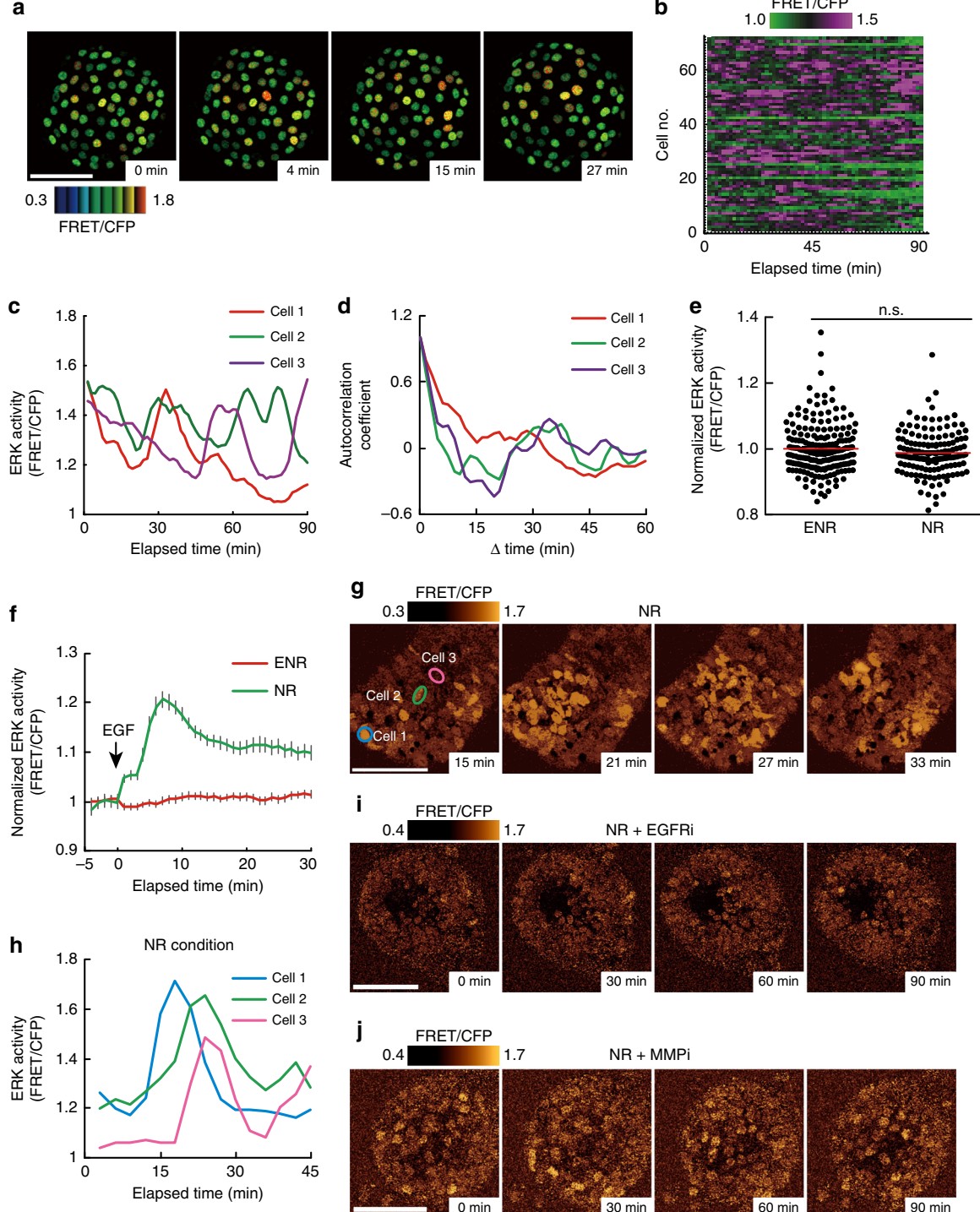

**Fig. 3** Spontaneous pulse-like ERK activation and propagation of ERK activity in intestinal organoids. **a–d** Time-lapse imaging of the EKAREV-NLS organoids was performed for 90 min at 1.5 min intervals ($n = 71$ cells). **a** Representative time-lapse images of ERK activity (FRET/CFP ratio) in an organoid showing pulse-like ERK activation. **b** Heat map showing the time course of ERK activity (FRET/CFP ratio) in each cell. **c** Time courses of ERK activity in three representative cells. The original data before smoothing by the moving average are shown in Supplementary Fig. 2c. **d** Temporal autocorrelation coefficients of ERK activity in the three cells shown in **c**. **e**, **f** Time-lapse imaging of the EKAREV-NLS organoids cultured under EGF-starved conditions for 24 h. **e** Bee swarm plots of ERK activity in organoids cultured under the normal condition (ENR) or EGF-starved condition (NR) ($n = 193$ and 148 cells pooled from three organoids). **f** Time course of the average ERK activity in the control or EGF-starved organoids after stimulation with EGF. The organoids were cultured under the normal (ENR) or EGF-starved condition (NR), and then stimulated with 50 ng ml⁻¹ of EGF at time point 0 (ENR: $n = 129$, NR: $n = 59$ cells). **g** Representative time-lapse images of an organoid cultured under the NR condition showing propagation of ERK activity. ERK activity (FRET/CFP ratio) is shown in golden pseudo colour mode. **h** Time courses of ERK activity in the three representative cells marked in **g**. **i**, **j** Representative time-lapse images of organoids treated with an EGFR inhibitor (PD153035, 1 μM) (**i**) or an MMP inhibitor (marimastat, 100 μM) (**j**). Organoids were cultured in NR media for 24 h, and subsequently treated with either inhibitor. ERK activity is shown in the golden pseudo colour mode. Scale bars, 50 μm. Red lines represent mean. Error bars represent s.e.m. Mann–Whitney $U$-test was used for comparison (**e**). *$P < 0.05$, **$P < 0.001$, ***$P < 0.0001$, n.s., not significant

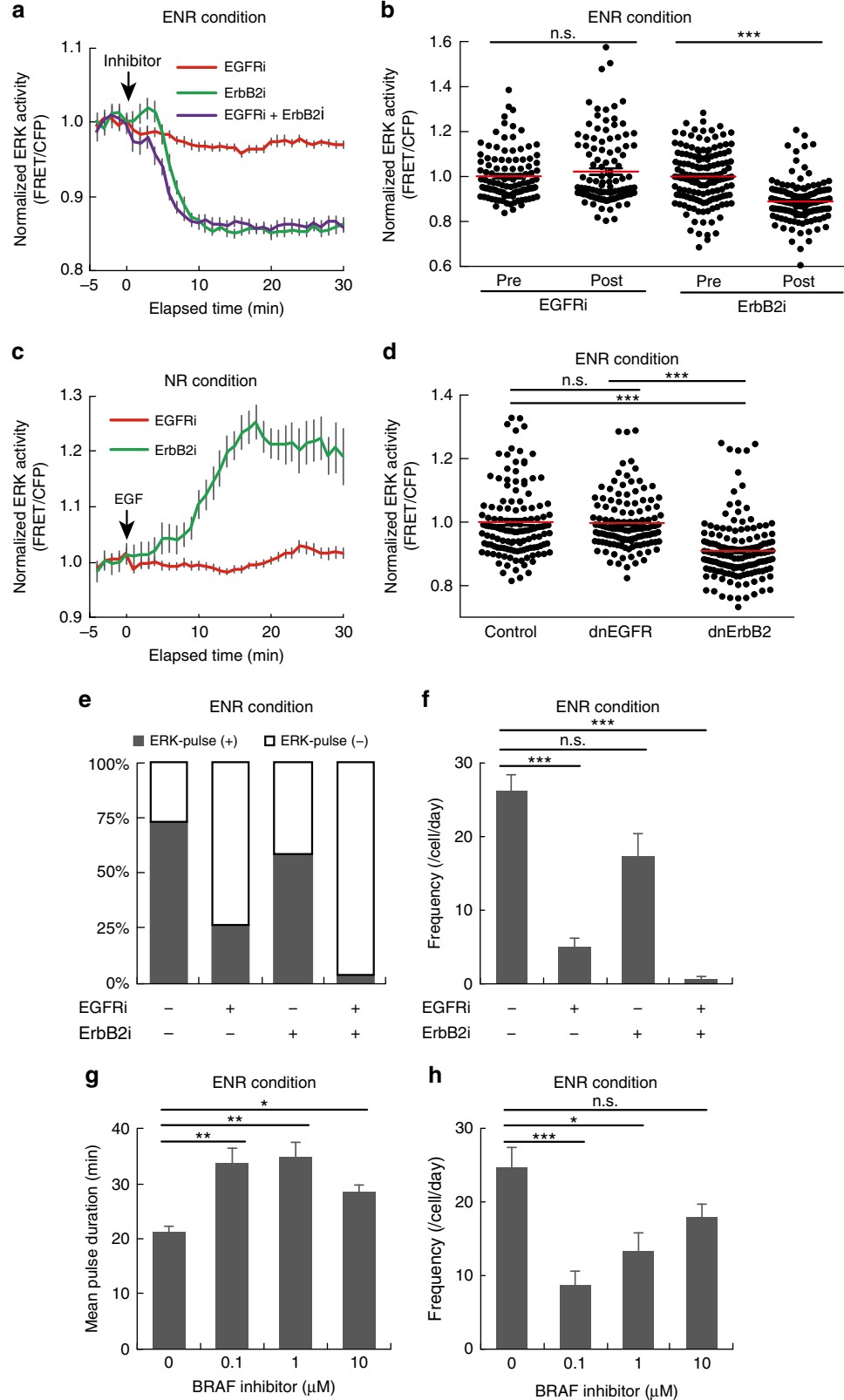

cancer therapy[3, 27, 39], we examined the effect of an EGFR inhibitor, PD153035, and an ErbB2 inhibitor, CP-724714, on ERK activity dynamics. Basal ERK activity was markedly decreased by CP-724714 (ErbB2i), but not by PD153035 (EGFRi) (Fig. 4 a, b). As expected, EGFRi but not ErbB2i abrogated EGF-induced ERK activation (Fig. 4c), validating the specificity of the inhibitors. To

confirm the effects of EGFR and ErbB2 inhibition on basal ERK activity, dominant negative forms of EGFR (dnEGFR) and ErbB2 (dnErbB2), both of which are truncated mutants lacking their respective intracellular domains[40, 41], were introduced into the intestinal organoids by using lentiviruses. Expression of dnErbB2, but not that of dnEGFR, significantly decreased the basal ERK

**Fig. 4** EGFR and ErbB2 generate two distinct modes of ERK activity in intestinal organoids. **a, b** EKAREV-NLS organoids were treated with EGFRi (PD153035) (1 μM), and/or ErbB2i (CP-724714) (10 μM), at time point 0. **a** Time courses of average ERK activity under each condition (EGFRi: $n = 70$, ErbB2i: $n = 62$, EGFRi + ErbB2i: $n = 57$ cells). **b** Quantification of ERK activity before and after treatment with EGFRi or ErbB2i (EGFRi: $n = 113$, ErbB2i: $n = 147$ cells, pooled from two organoids). **c** Time course of average ERK activity in organoids that were cultured under EGF-starved condition (NR) for 24 h, treated with either EGFRi or ErbB2i for 60 min, and then stimulated with 50 ng ml$^{-1}$ of EGF (EGFRi: $n = 48$, ErbB2i: $n = 32$ cells). **d** Quantification of ERK activity in organoids infected with a control lentivirus (control) or lentiviruses expressing either a dominant negative form of EGFR (dnEGFR) or that of ErbB2 (dnErbB2) (Control: $n = 134$, dnEGFR: $n = 129$, dnErbB2: $n = 151$ cells, pooled from five organoids cultured in the ENR medium). **e, f** Quantification of ERK activity pulses in EKAREV-NLS organoids cultured in the ENR medium and treated with EGFRi and/or ErbB2i (−/−: $n = 71$, EGFRi/−: $n = 49$, −/ErbB2i: $n = 36$, EGFRi/ ErbB2i: $n = 53$ cells). Organoids were treated with indicated inhibitors and imaged for 90 min. ERK activity data were smoothened by 6-min moving average, and fitted to flat lines or multi-peak functions. **e** The proportion of cells exhibiting the pulse-like ERK activation (ERK-pulse$^+$) under each condition. **f** Frequencies of ERK activity pulses under each condition. Duration (**g**) and frequencies (**h**) of ERK activity pulses in organoids cultured in the ENR medium and treated with 0, 10, 1, or 0.1 μM of a BRAF inhibitor (SB590885) for 90 min ($n = 55$, 52, 29, and 65 cells, respectively). Scale bars, 50 μm. Red lines represent mean. Error bars represent s.e.m. Mann–Whitney $U$-test (**b**) and Steel–Dwass test (**d, f–h**) were used for comparison. *$P < 0.05$, ** $P < 0.001$, ***$P < 0.0001$, n.s., not significant

---

activity ($P < 0.0001$) (Fig. 4d). Moreover, knockdown of ErbB2 or EGFR by short hairpin RNA exerted similar effects on the basal ERK activity (Supplementary Fig. 3a, b). Taken together, these results show that ErbB2, but not EGFR, mainly drives the basal ERK activity in intestinal organoids.

We then examined the role of EGFR and ErbB2 in generating the ERK activity pulses. To extract the quantitative parameters of the pulses, time-course data of ERK activity in each cell were fitted to horizontal lines (corresponding to the sustained, constant activity) or multi-peak functions (for the pulse-like activity). We categorized cells exhibiting one or more ERK activity pulses, as ERK-pulse$^+$, or otherwise as ERK-pulse$^-$ (Supplementary Fig. 3c–e). Treatment with EGFRi, but not that with ErbB2i, decreased the proportion of ERK-pulse$^+$ cells (Fig. 4e). Moreover, the frequency of ERK activity pulses was substantially reduced by EGFRi, but not by ErbB2i (Fig. 4f). Upon co-inhibition of EGFR and ErbB2, the ERK activity pulses disappeared almost completely (Fig. 4e, f). In line with this, knockdown of EGFR, but not that of ErbB2, reduced the frequency of ERK activity pulses (Supplementary Fig. 3f). These results show that, in intestinal organoids, ERK activity pulses are generated by EGFR kinase activity whereas basal ERK activity is maintained by ErbB2 kinase activity. Removal of R-spondin 1 or Noggin did not affect basal ERK activity, its dependency on ErbB2, and the dependency of ERK activity pulses on EGFR (Supplementary Fig. 4a–f), negating involvement of these proteins in controlling ERK activity dynamics in intestinal organoids. Interestingly, treatment with a BRAF inhibitor, SB590885, not only decreased the frequency but also prolonged the duration of ERK activity pulses (Fig. 4g, h). As the shape of pulses generally depends on the activity of the pulse generator, BRAF might be one of the pulse generators in this system.

**EGFR governs distinct ERK activity dynamics in adenoma cells**. The above results demonstrated distinct roles for EGFR and ErbB2 in regulating the ERK activity dynamics in normal cells. We next investigated how these receptors regulate ERK activity dynamics during intestinal tumorigenesis. Organoids were generated from adenomas developed in Apc$^{Δ716}$ mice, which have a truncation mutation in Apc[42], followed by introduction of the biosensor for ERK. Under our experimental conditions, approximately half of the cells in the adenoma-derived organoids expressed the biosensor (Fig. 5a). There was no significant difference in basal ERK activity between the normal and adenoma-derived organoids ($P = 0.7599$) (Fig. 5a, b). However, a striking difference was observed in the presence of EGFRi and ErbB2i: treatment with EGFRi, which did not affect basal ERK activity in the normal organoids (Fig. 4a, b), significantly decreased it in the adenoma-derived organoids ($P < 0.0001$)

(Fig. 5c, d). Moreover, adenoma-derived organoids were less sensitive to ErbB2i compared to the normal organoids (Figs. 5d and 4b). Meanwhile, EGFRi, but not ErbB2i, decreased the frequency of the ERK activity pulses (Fig. 5e) and the proportion of ERK-pulse$^+$ cells in adenoma-derived organoids (Fig. 5f). Treatment with both inhibitors almost completely suppressed the firing of ERK activity pulses (Fig. 5e, f). Altogether, these results indicate that EGFR signalling is a predominant driver for basal ERK activity and spontaneous ERK activity pulses in adenoma-derived organoids.

**Wnt signalling activation alters ERK activity dynamics**. Although deregulated/constitutive activation of Wnt signalling caused by APC mutations is known as the initial genetic event in the adenoma-carcinoma sequence[20], accumulating evidences indicate that tumour cells have already acquired epigenetic alterations before the APC mutations[43,44]. Therefore, we inquired whether activation of Wnt signalling alone could recapitulate the altered ERK activity dynamics in adenoma-derived organoids. We utilized a GSK3 inhibitor, CHIR99021, which causes β-catenin accumulation thereby activating canonical Wnt/β-catenin signalling. CHIR99021 did not affect the basal ERK activity at least for 24 h (Fig. 6a, b); however, CHIR99021 surprisingly changed the cellular responses to EGFRi and ErbB2i. In organoids treated with CHIR99021, EGFRi decreased the basal ERK activity more potently than did ErbB2i (Fig. 6c), as observed in the adenoma-derived organoids. Simultaneous addition of EGFRi and ErbB2i most strongly suppressed the basal ERK activity (Fig. 6c). The effects of CHIR99021 disappeared after 24 h incubation in ENR medium (without CHIR99021), suggesting that CHIR99021-induced sensitivity to EGFRi was reversible (Fig. 6d). CHIR99021 also increased the proportion of cells with ERK activity pulses (ERK-pulse$^+$) (Fig. 6e) and the frequency of the pulses (Fig. 6f). The firing of the ERK activity pulse was suppressed almost completely by EGFRi and slightly by ErbB2i (Fig. 6e, f). Notably, Wnt ligand stimulation or overexpression of constitutively active β-catenin exerted effects similar to those of CHIR99021 on ERK activity dynamics (Supplementary Fig. 5). These results suggest that activation of Wnt signalling increases the contribution of EGFR signalling to ERK activity thereby altering the ERK activity dynamics similar to those in adenoma-derived organoids.

To examine the biological significance of Wnt signalling-dependent alterations in ERK activity dynamics, we conducted cell cycle analyses in intestinal organoids. We generated intestinal organoids from transgenic mice expressing a cell cycle reporter (Fucci2a), which marks S/G2/M phase cells and G1 phase cells by mVenus and mCherry, respectively[45] (Fig. 6g). Consistent with an essential role of ERK in cell proliferation, MEK inhibitor treatment strongly reduced the proportion of S/G2/M phase cells

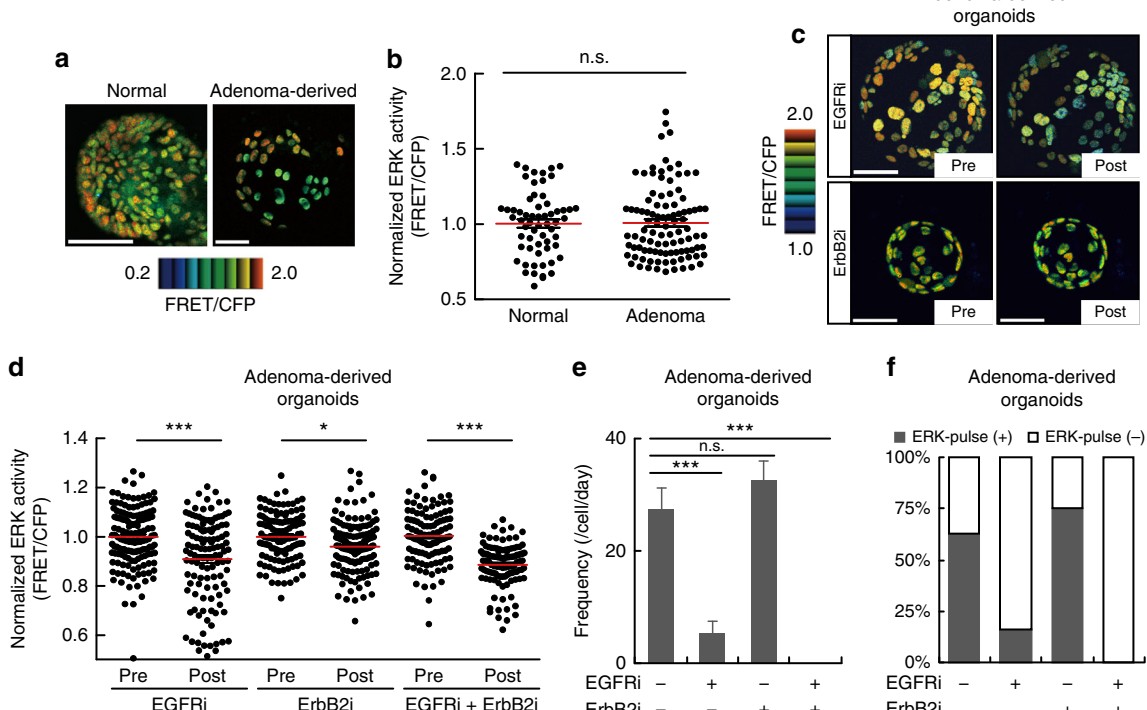

**Fig. 5** Adenoma-derived organoids exhibit increased dependence on EGFR signalling. **a**, **b** Organoids were generated from the normal epithelium or adenomas of the Apc$^{\Delta 716}$ mouse small intestine, and infected with lentiviruses expressing a FRET biosensor for ERK activity (EKAREV-NLS). **a** Representative images of ERK activity (FRET/CFP ratio) in organoids derived from the normal epithelium and adenomas. **b** Bee swarm plots of ERK activity in the normal epithelium-derived and adenoma-derived organoids (Normal: $n = 61$, Adenoma: $n = 99$ cells, from more than three organoids). **c**–**f** Adenoma-derived organoids expressing EKAREV-NLS were treated with 1 μM of an EGFR inhibitor (EGFRi), PD153035, and/or 10 μM of an ErbB2 inhibitor (ErbB2i), CP-724714. **c** Representative images of adenoma-derived organoids before and after treatment with EGFRi or ErbB2i. **d** Bee swarm plots of ERK activity before and after inhibitor treatment (EGFRi: $n = 120$, ErbB2i: $n = 119$, EGFRi + ErbB2i: $n = 109$ cells, pooled from three organoids). **e**, **f** Quantification of ERK activity pulses in adenoma-derived organoids. Organoids were treated with EGFR and/or ErbB2 inhibitors, and then imaged for 90 min. ERK activity data from each cell were processed as described for Fig. 4e (−/−: $n = 40$, EGFRi/−: $n = 45$, −/ErbB2i: $n = 32$, EGFRi/ErbB2i: $n = 29$ cells). Frequencies of the pulse-like ERK activity (**e**) and the proportion of cells with pulse-like ERK activity (ERK-pulse$^+$) (**f**) under each condition. Scale bars, 50 μm. Red lines represent mean. Error bars represent s.e.m. Mann–Whitney $U$-test (**b**, **d**) and Steel–Dwass test (**e**) were used for comparison. *$P < 0.05$, **$P < 0.001$, ***$P < 0.0001$, n.s., not significant

(Fig. 6h). ErbB2i, but not EGF depletion or EGFRi, significantly decreased the proportion of S/G2/M phase cells ($P < 0.0001$) (Fig. 6h). The effect of ErbB2i was further enhanced by simultaneous addition of EGFRi (Fig. 6h). As expected, removal of R-spondin 1 or Noggin also decreased S/G2/M phase cells, consistent with their roles in maintaining undifferentiated IECs (Supplementary Fig. 6a, b). Even under the R-spondin 1- or Noggin-depleted condition, ErbB2i reduced S/G2/M phase cells more strongly than EGFRi (Supplementary Fig. 6a, b). These results suggest that ErbB2 primarily drives cell cycle progression in normal intestinal organoids and that EGFR serves as the auxiliary driver. Thus, basal ERK activity driven by ErbB2 signalling might play a major role in cell proliferation in the normal organoids. Notably, treatment with CHIR99021 or Wnt ligands, which increased the frequency of ERK activity pulses without affecting the basal ERK activity (Fig. 6f, Supplementary Fig. 5c), increased proliferating cells in an EGFR-dependent manner (Fig. 6i, Supplementary Fig. 6c–e). Thus, the increased frequency of ERK activity pulses correlated with increased cell proliferation after those treatments.

**GSK3 inhibitor treatment enhances EGFR signalling in vivo**. The results from intestinal organoid studies suggest the Wnt signalling-dependent control of EGFR signalling in IECs. We thus inquired whether administration of a GSK3 inhibitor affects EGFR signalling in the mouse intestinal epithelium, as does it in intestinal organoids. To this end, CHIR99021 was administered to EKAREV-NLS mice, and ERK activity before and after administration of a clinically used EGFR inhibitor, erlotinib was observed. Administration of CHIR99021 increased the expression of several genes, which were also increased by CHIR99021 in intestinal organoids and have been reported as the Wnt-target genes, in the intestinal epithelium (Supplementary Fig. 7a). Intravital imaging demonstrated that erlotinib decreased the basal ERK activity in the intestinal epithelium of CHIR99021-treated mice but not in that of control mice (Fig. 7a). Notably, CHIR99021 increased the expression of EGFR protein in the crypt region that mainly comprised stem cells and progenitor cells (Fig. 7b, c). The EGFR protein was expressed more abundantly in the crypt than in the villus (Fig. 7b), where Wnt signalling activity should be low[46]. Moreover, EGFR expression was markedly increased in adenomas of Apc$^{\Delta 716}$ mice (Fig. 7d, e), as well as adenoma-derived organoids (Fig. 7f, g) and CHIR99021-treated organoids (Supplementary Fig. 7b, c). Thus, the increased expression of EGFR might be involved in the Wnt signalling-dependent augmentation of EGFR signalling. The expression levels of ErbB2 were comparable among adenomas and the normal epithelium (Supplementary Fig. 7d). Finally, we examined whether alteration in EGFR signalling activity affects cell

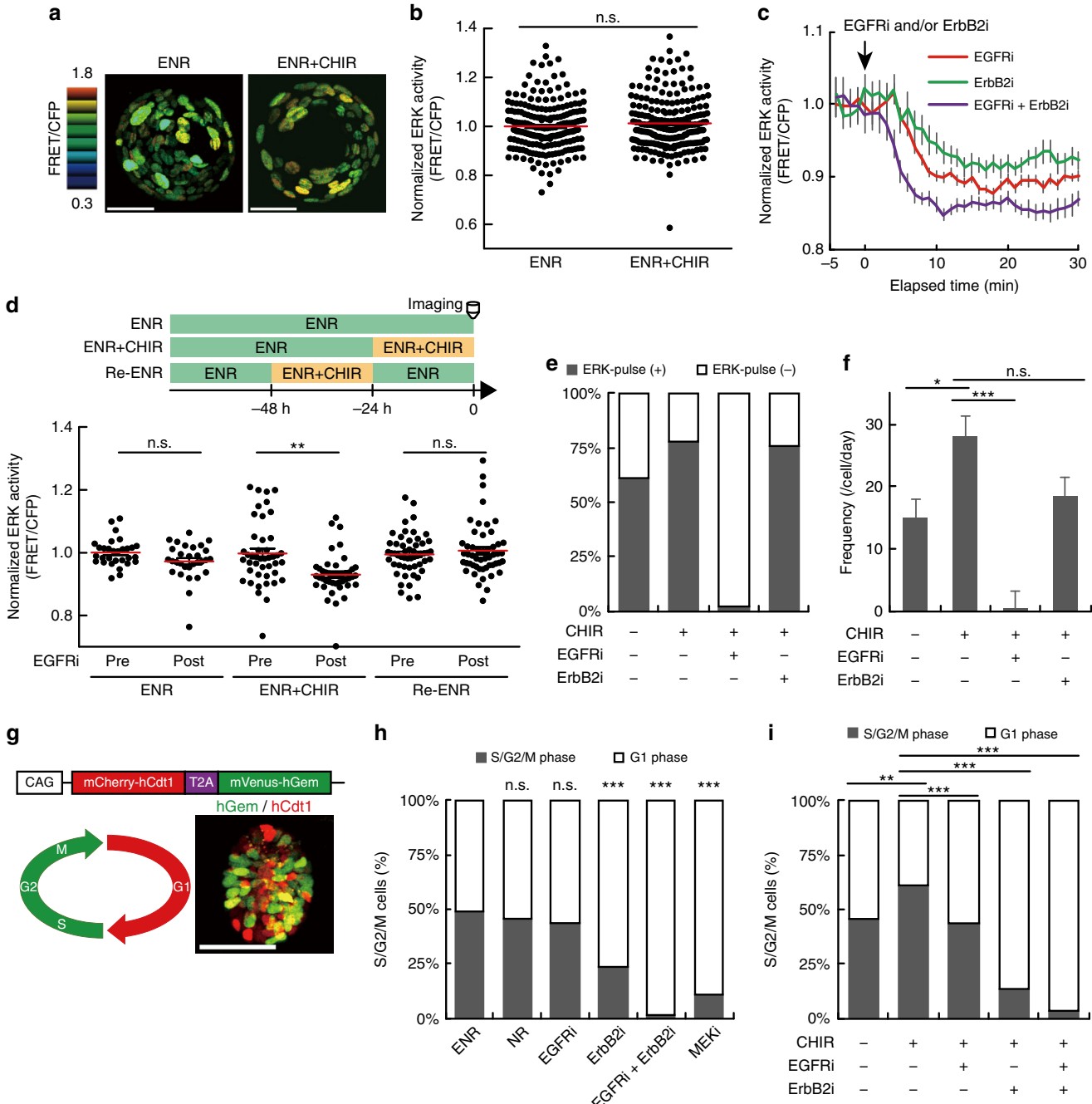

**Fig. 6** Pharmacological activation of Wnt signalling promotes cell proliferation by augmenting EGFR signalling. **a** Representative images and **b** quantification of ERK activity in organoids treated with or without CHIR99021 (5 μM) for 24 h ($n = 210$ and 213 cells (from the left), from three organoids). **c** Time courses of ERK activity in organoids treated with CHIR99021 for 24 h and then treated with EGFRi (PD153035), and/or ErbB2i (CP-724714) ($n = 45$, 29, and 29 cells from the left). **d** Quantification of ERK activity before (Pre) and after (Post) EGFRi treatment (bottom) ($n = 30$, 44, and 53 cells from the left). Organoids cultured in ENR, ENR+CHIR, or ENR+CHIR99021 for 24 h and then in ENR for another 24 h (Re-ENR) were treated with EGFRi (top). **e**, **f** Quantification of ERK activity pulses in organoids treated with CHIR99021 for 24 h and then with EGFRi and/or ErbB2i for 90 min during imaging. The proportion of cells with ERK activity pulses (ERK-pulse[+]) (**e**), and the frequency of the pulses (**f**) under each condition ($n = 36$, 47, 47, and 25 cells from the left). **g** Schematic structure of Fucci2a (top), cell cycle diagram marked with corresponding fluorescence (bottom left), and the representative image of organoid expressing Fucci2a (bottom right). **h**, **i** Proportion of cells in the S/G2/M phases in the Fucci2a organoids. **h** Organoids were cultured for 24 h in ENR, EGF-deficient medium (NR), or ENR supplemented with EGFRi ErbB2i, and/or a MEK inhibitor (MEKi) ($n = 158$, 264, 400, 359, 423, and 243 cells (from the left), from more than four organoids). **i** Organoids treated with CHIR99021 for 24 h were subsequently treated with EGFRi, and/or ErbB2i for another 24 h ($n = 156$, 1062, 461, 220, and 234 cells (from the left), from more than four organoids). Scale bars, 50 μm. Red lines represent mean. Error bars represent s.e.m. Mann–Whitney $U$-test (**b**, **d**), Steel–Dwass test (**f**), and $\chi^2$ test with BH procedure (**h**, **i**) were used for comparison. *$P < 0.05$, **$P < 0.001$, ***$P < 0.0001$, n.s., not significant

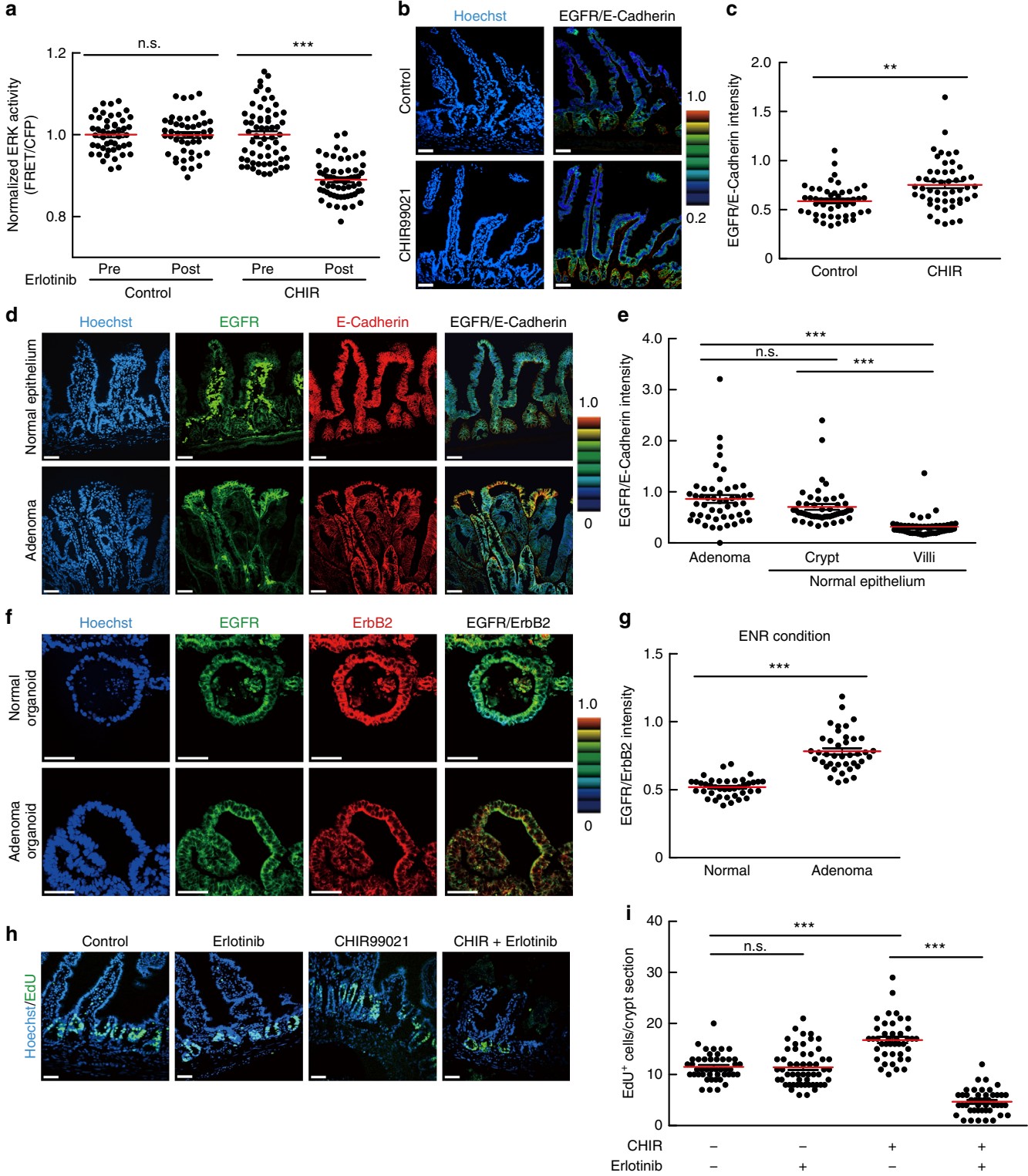

proliferation and survival in the intestinal epithelium. We found that administration of CHIR99021 promoted proliferation of crypt cells, whereas erlotinib did not exhibit a significant effect (Fig. 7h, i, Supplementary Fig. 7e, f). Remarkably, simultaneous administration of CHIR99021 and erlotinib strongly suppressed the proliferation of crypt cells (Fig. 7h, i, Supplementary Fig. 7e, f), promoted apoptosis (Supplementary Fig. 7g), and severely disturbed the crypt–villus structures (Fig. 7h). These results suggest that in vivo administration of CHIR99021 enhances EGFR

signalling and promotes proliferation and survival of IECs, and that enhancement of EGFR signalling renders cells highly sensitive to EGFR inhibition in vivo. It is also noteworthy that autophosphorylation of ErbB2 was suppressed by ErbB2i, but not by EGFRi, in normal and adenoma organoids (Supplementary Fig. 8a–d). Thus, EGFR might be dispensable for ErbB2 activation.

**Wnt signalling controls ERK activity via EGFR regulators.** Finally, we examined whether Wnt signalling-induced gene

**Fig. 7** EGFR signalling is augmented in adenomas and CHIR99021-treated mouse intestine in vivo. **a** Bee swarm plots of ERK activity in the vehicle- (Control) or CHIR99021-treated (CHIR) mouse intestinal epithelium before (Pre) and after (Post) treatment with an EGFR inhibitor, erlotinib. Mice were injected with vehicle or 20 mg kg$^{-1}$ body weight of CHIR99021 for 3 days and then treated with 100 mg kg$^{-1}$ body weight erlotinib for 30 min (Control: $n =$ 48, CHIR: $n =$ 61 cells from three crypts). **b**, **c** Immunofluorescence staining of the small intestine in the mice injected with vehicle (Control) or CHIR99021 for 3 days using anti-EGFR and anti-E-cadherin antibodies. **b** The ratio of EGFR and E-cadherin staining intensities is shown in the IMD mode according to the colour scale. Counterstaining was performed with Hoechst. Note that strong staining of stromal cells results from non-specific binding of the secondary antibodies used here (anti-mouse IgG). **c** Quantification of the EGFR/E-cadherin staining intensity ratio in each cell under vehicle or CHIR99021-treated condition ($n =$ 50 cells). **d** Immunofluorescence staining of an adenoma and the normal small intestine of Apc$^{\Delta716}$ mice with anti-EGFR and anti-E-cadherin antibodies. EGFR/E-cadherin staining intensity ratio is shown in the IMD mode. **e** Quantification of the EGFR/E-cadherin ratio in each cell located in the indicated regions ($n =$ 50 cells). **f** Immunofluorescence staining of normal and adenoma-derived organoids with anti-EGFR and anti-ErbB2 antibodies. EGFR/ErbB2 staining intensity ratio is shown in the IMD mode. **g** Quantification of the EGFR/ErbB2 ratio in each cell in the organoids ($n =$ 40 cells pooled from at least two organoids). **h**, **i** EdU staining of the small intestine of CHIR99021- and/or erlotinib-treated mice. Mice were injected with vehicle, 20 mg kg$^{-1}$ body weight of CHIR99021, and/or 100 mg kg$^{-1}$ body weight of erlotinib for 3 days. **i** Quantification of EdU$^+$ cells per crypt section ($-/-$: $n =$ 46, CHIR/$-$: $n =$ 42, $-$/Erlotinib: $n =$ 54, and CHIR/Erlotinib: $n =$ 45 crypts from three mice). Scale bars, 50 μm. Red lines represent mean. Error bars represent s.e.m. Mann–Whitney $U$-test (**a**, **c**, **g**), and Steel–Dwass test (**e**, **i**) were used for comparison. *$P < 0.05$, **$P < 0.001$, ***$P < 0.0001$

expression changes regulate EGFR–ERK signalling dynamics. For this purpose, we analyzed gene expression profiles of normal (ENR) and CHIR99021-treated organoids (ENR + CHIR) by using microarrays. We identified 81 and 274 transcripts whose expression levels were upregulated and downregulated, respectively, in CHIR99021-treated organoids by more than 3 folds (Fig. 8a, Supplementary Data 1). Gene set enrichment analysis revealed that CHIR99021 induced gene expression changes similar to those in the *Apc*-knockout epithelium and colorectal adenomas (Fig. 8b–e). Indeed, gene sets comprising genes downregulated and upregulated after *Apc* knockout were most significantly enriched in the CHIR99021-dependent down-regulated and upregulated gene list, respectively (Supplementary Table 1). Notably, CHIR99021 altered the expression of four genes whose protein products promote (Egfl6, Flna, and Troy) or suppress (Lrig3) EGFR signalling[47–50] (Supplementary Data 1). The expression levels of Dusps, Sprys, Spreds, and MIG6, which also regulate EGFR–ERK signalling[35], were not significantly altered by CHIR99021 (Supplementary Fig. 9a). RT-PCR analyses confirmed that Egfl6, Flna, and Troy were upregulated, and that Lrig3 was downregulated in CHIR99021-treated organoids (Fig. 8f) and adenoma-derived organoids (Fig. 8g). To examine whether these genes mediate the altered ERK activity dynamics in adenomas, we performed exogenous expression and knockdown of these genes in adenoma-derived organoids (Supplementary Fig. 9b). The knockdown of Egfl6 or expression of Lrig3 abolished the sensitivity of adenoma cells to EGFR inhibition (Fig. 8h). Meanwhile, the frequency of ERK activity pulses was significantly decreased by the Lrig3 expression ($P = 0.0001$) and the Troy knockdown ($P = 0.0009$) in adenoma organoids (Fig. 8i). Knockdown of Flna did not significantly affect ERK activity dynamics in either experiment (Fig. 8h, i). These results suggest that decreased expression of Lrig3 and increased expression of Egfl6 and Troy, probably together with enhanced EGFR expression, coordinately promotes EGFR–ERK signalling in adenoma cells.

## Discussion

The spatiotemporal regulation of ERK signalling has been considered a key determinant of cellular responses in many biological contexts[51, 52], and its disturbance has been implicated in many diseases including cancers[2]. Here, we successfully visualized ERK activity in the intestinal epithelium and unveiled the roles of EGFR and ErbB2 in regulating ERK activity dynamics. We found that EGFR kinase activity is required for ERK activity pulses, and that ErbB2 is the primary tyrosine kinase governing basal ERK activity and cell proliferation in intestinal organoids (Figs. 4 and 6). ERK activity in IECs comprises the ErbB2-dependent

constant component and EGFR-dependent pulse-like component (Fig. 8j). Remarkably, Wnt signalling activation, which often occurs during intestinal tumorigenesis, enhanced EGFR–ERK signalling, at least in part, by controlling expression of EGFR and its regulators, such as Egfl6, Lrig3, and Troy (Figs. 7 and 8). This alteration increased the contribution of EGFR to the basal ERK activity, increased the frequency of ERK activity pulses, and promoted proliferation of IECs (Figs. 6 and 7). Thus, Wnt signalling activation sensitized IECs to pharmacological inhibition of EGFR signalling (Fig. 7). These results show that distinct upstream tyrosine kinase receptors can generate different modes of ERK activity dynamics to control cellular responses, and that deregulation of such mechanisms might underlie the characteristic features of cancer cells.

Upon Wnt signalling activation, augmented EGFR signalling increased the frequency of ERK activity pulses, which was correlated with the increased proliferation of IEC (Figs. 6 and 7). Importantly, however, these results do not indicate that the pulsatile nature of EGFR–ERK signalling per se is needed for the promotion of cell proliferation. Notably, ErbB2 inhibition decreased the basal ERK activity less efficiently in adenomas or in GSK3 inhibitor-treated organoids than in normal organoids (Figs. 5 and 6). This shows that, after Wnt signalling activation, basal ERK activity is maintained at a constant level through adaptation to enhanced EGFR signalling, which increases the relative contribution of EGFR to the ERK activity.

We demonstrated that EGFR, but not ErbB2, is a major generator of the ERK activity pulses (Fig. 4e, f). The distinct contribution of EGFR and ErbB2 to ERK activity dynamics might be attributed to the different regulatory mechanisms involved. Since EGFR–ERK signalling activity can be affected by the fluctuation of the topical concentration of EGFR ligands and also be regulated by multiple negative feedback mechanisms[35, 36, 53], these effects might introduce a pulsatile nature into EGFR signalling. In contrast to EGFR, ErbB2 does not have any known ligands and its activity is mainly regulated by dimerization[54]. In addition, ErbB2-containing heterodimers, compared to dimers without ErbB2, exhibit slower rates of endocytosis and dissociation from ligands[55–57]. Considering all these studies, the ErbB2 homodimer and ErbB2-containing heterodimers might be more stable and transmit sustained, constant inputs to ERK. Importantly, knockdown of EGFR did not affect basal ERK activity (Supplementary Fig. 3b), which is dependent on ErbB2 kinase activity. Moreover, an ErbB2 inhibitor, but not an EGFR inhibitor, reduced autophosphorylation of ErbB2 (Supplementary Fig. 8a–d), suggesting that the majority of ErbB2 autophosphorylation is mediated by ErbB2, but not by EGFR. Thus, ErbB2 seems to be activated mainly via homodimerization,

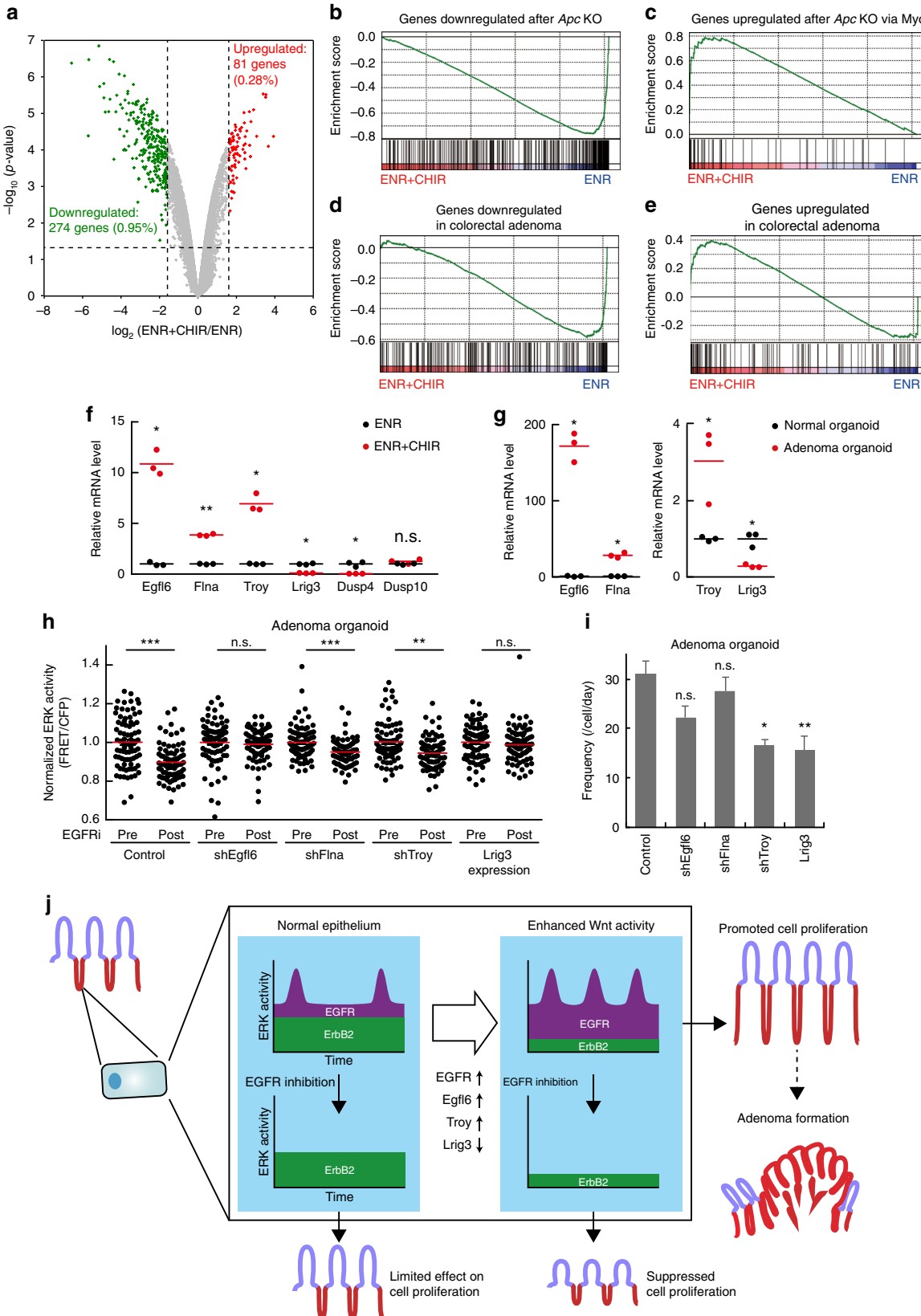

although our data could not rule out the involvement of EGFR and ErbB3 in the ErbB2 activation. In any case, our results indicate that the upstream receptors under the control of different regulatory mechanisms can generate distinct modes of ERK activity dynamics. Consistently, distinct mechanisms acting at the

receptor level are involved in different ERK activity dynamics in cultured mammary epithelial cells[58]. Since our results also suggest the involvement of Raf in generating ERK activity pulses (Fig. 4g, h), multiple feedback loops might govern the pulses at different levels of the signalling cascade. It should be clarified that EGFR

**Fig. 8** Wnt signalling activation affects EGFR–ERK signalling dynamics through regulating expression of multiple molecules. **a** A volcano plot depicting the fold changes in gene expression levels between normal (ENR) and CHIR99021-treated organoids (ENR + CHIR), and statistical significance of the changes. **b**–**e** Enrichment plots from gene set enrichment analysis (GSEA). GESA plots for genes downregulated after *Apc* knockout (**b**), genes upregulated after *Apc* knockout through *Myc* (**c**), genes downregulated in colorectal adenoma (**d**), and genes upregulated in colorectal adenoma (**e**) are shown. **f**, **g** RT-PCR analysis revealed that Egfl6, Flna, and Troy were upregulated, and that Lrig3 was downregulated in both CHIR99021-treated and adenoma-derived organoids. The relative mRNA levels of indicated genes in normal (ENR) versus CHIR99021-treated organoids (ENR + CHIR) (**f**), and those in normal versus adenoma-derived organoids (**g**) are shown (*n* = 3 samples containing more than ten organoids). **h**, **i** Adenoma-derived organoids expressing the ERK biosensor were infected with lentiviruses expressing control vector (control), Lrig3 (Lrig3), or shRNAs for Egfl6 (shEgfl6), Flna (shFlna), or Troy (shTroy). **h** Bee swarm plots of ERK activity in organoids before (Pre) and after (Post) EGFR inhibitor treatment under each condition (*n* = 80 cells pooled from two organoids). **i** The frequency of ERK activity pulses under each condition. Time-lapse imaging was performed for 90 min (*n* = 50 cells). **j** Schematic representation of ERK activity dynamics generated by kinase activity of EGFR and ErbB2 in the normal and Wnt signalling-activated intestinal epithelia. Red lines represent mean. Error bars represent s.e.m. Welch's *t* test (**f**, **g**), Mann–Whitney *U*-test (**h**), and Steel–Dwass test (**i**) were used for comparison.*P < 0.05, **P < 0.001, ***P < 0.0001

has a kinase activity-independent function as a heterodimerization partner of ErbB2. However, our results did not support the contribution of EGFR to ErbB2 activation, as discussed above. Thus, the observed functions of EGFR are likely to be mediated by intracellular signalling triggered by EGFR kinase activity, which is referred to as EGFR signalling in this study.

Different dynamics of receptor tyrosine kinases could lead to distinct dynamics of the downstream ERK activity[35]. For example, in PC12 cells, activated EGFR rapidly undergoes the clathrin-based endocytosis and the following degradation, whereas activation of TrkA, a receptor for NGF, induces its translocation to long-lived signalling endosomes, where TrkA avoids degradation and mediates sustained ERK activation[59]. Thus, mechanisms regulating dynamics of specific receptors can regulate ERK activity dynamics. In this study, we identified Egfl6, Lrig3, and Troy as important regulators of EGFR–ERK signalling dynamics in IECs (Fig. 8). Previously, these proteins have been reported to regulate EGFR signalling in cultured cells[47–49]. Notably, altered expression of these regulators was associated with changes in ERK activity dynamics in adenoma cells (Fig. 8). Thus, elucidation of mechanisms by which these regulators function in IECs would provide a clue about how ERK activity dynamics can be altered during tumorigenesis and how it affects tumour cell phenotypes.

EGFR signalling has been implicated in many cancers and its pharmacological inhibition has been considered promising for cancer treatment[4, 5]. Our results might provide a therapeutic rationale for why EGFR inhibitors can specifically suppress the growth of certain types of CRCs without severe adverse effects. Since strong activation of Wnt signalling augments EGFR signalling and renders cells highly dependent on EGFR signalling (Figs. 6 and 7), it can be readily speculated that pharmacological inhibition of EGFR preferentially targets CRCs rather than normal IECs harbouring weaker Wnt signalling activity. Consistently, administration of a GSK3 inhibitor (CHIR99021), which promoted expression of, at least a fraction of, Wnt-target genes, increased the sensitivity of normal IECs to EGFR inhibition in vivo (Fig. 7). Importantly, however, the effects of CHIR99021 administration and Wnt signalling activation are not completely the same, as many events other than GSK3β inhibition occur in Wnt signalling and GSK3 plays various roles independently of Wnt signalling. Also, the effects of CHIR99021 were not as strong as those of Wnt signalling activation; CHIR99021 promoted cell proliferation only weakly whereas genetic activation of Wnt signalling has been shown to cause hyperplasia in the intestinal epithelium. Nevertheless, CHIR99021 exerted similar effects on ERK activity dynamics and expression of several genes both in vitro and in vivo. In addition to EGFR signalling, our results also demonstrate the significant contribution of ErbB2 signalling to ERK activity in IECs. Indeed, proliferation of IECs was more effectively suppressed by concomitant inhibition of EGFR and

ErbB2compared to inhibition of either one. Consistent with this finding, simultaneous inhibition of EGFR and ErbB2 has been shown to be effective in refractory CRCs[60], though inhibition of ErbB2 alone was ineffective. Thus, our findings and the proposed model are in good agreement with the clinical efficacy of EGFR and ErbB2 inhibition in CRC treatment. Nevertheless, it should be noted that there are significant differences between mouse adenomas and human CRCs. In particular, mouse adenomas are usually assumed to have only *Apc* mutations, whereas human CRCs often contain many genetic mutations, which might affect cellular responses to EGFR inhibition. Indeed, CRCs harbouring *RAS* or *RAF* mutations are resistant to EGFR inhibitors, and initially sensitive CRCs can also acquire the resistance via several mechanisms[28].

Recent studies have shown that ErbB signalling regulates the transition between quiescence and proliferation of ISCs[18, 19]. For instance, a pan-ErbB negative regulator, Lrig1, is expressed in a quiescent, long-lived subpopulation of ISCs, which is distinct from the Lgr5+ population and serves as an origin of intestinal tumours[18, 19, 61]. Lrig1 depletion increases the expression of ErbB family receptors and enhances ERK activity, resulting in ISC expansion and adenoma formation[18, 19]. Thus, Lrig1 plays a crucial role in controlling quiescence and proliferation of ISCs. In this study, we showed that Wnt signalling activation, increases the expression of EGFR, but not ErbB2 (Fig. 7). Since Lrig1 broadly regulates the expression of ErbB receptors[18, 19], selective induction of EGFR suggests Lrig1-independent regulation of EGFR expression. More recently, Basak and colleagues have reported that EGFR–ERK signalling mediates ISC proliferation and that inhibition of EGFR and ErbB2, MEK, or ERK induces reversible quiescence of ISCs[62]. Notably, they also showed that inhibition of EGFR alone could suppress proliferation of ISCs[62]. In contrast, we did not observe any significant alterations in cell proliferation after EGFR inhibition for 24 h. This discrepancy could be due to the difference in experimental settings, since they mainly focused on the effects of long-term perturbation on ISC functions[62] whereas we focused on the immediate effects of the treatment, which more likely reflect alterations in rapidly cycling progenitors rather than in ISCs. Since stem cells harbour strong Wnt signalling activity[46], inhibition of EGFR alone might suppress their proliferation in the long term.

In conclusion, we have revealed ERK activity dynamics regulated by two distinct upstream receptors, EGFR and ErbB2, in the intestinal epithelium and shown that alterations in the dynamics caused by Wnt signalling activation underlie the high sensitivity of tumour cells to EGFR inhibition. These findings highlight the importance of understanding ERK activity dynamics and their role in controlling cellular functions at the single cell level, issues that could not be addressed by conventional biochemical approaches. Similar phenomena could have occurred for many

other signalling pathways and molecules, elucidation of which should be an important future research area to improve the understanding of cellular drug responses at the single cell level. Live imaging with highly sensitive FRET biosensors could be a powerful tool to address these issues and shed new light on the mechanisms involved in many physiological and pathological processes.

## Methods

**Lentivirus production and infection of intestinal organoids**. The FRET biosensor for ERK has been described previously[7]. The dominant negative form of EGFR (dnEGFR) is a C-terminally truncated mutant encoding amino acid 1 to 679[40], and the dominant negative form of ErbB2 (dnErbB2) encodes 1 to 693, respectively[41]. Both were generated by PCR using mouse cDNA encoding the corresponding full-length protein as the template. Lentivirus expressing a constitutively active form of β-catenin was described previously[63]. Constructs expressing shRNAs were made by inserting corresponding oligonucleotides into a CSII-U6-MCS-puro vector[64]. The target sequences of shRNAs are shown in Supplementary Table 2. The FRET biosensor was cloned into pCSIIbsr, and dominant negative constructs were cloned into pCSIIpuro, both of which are derived from pCSII-EF (a gift from Hiroyuki Miyoshi)[65], to produce lentiviruses. For virus production, Lenti-X 293T cells (Clontech) were transfected with the pCSII plasmids, together with psPAX2 and pCMV-VSV-G-RSV-Rev plasmids. The culture supernatants containing the lentiviruses were collected 48 h after transfection, filtered, concentrated with the Lenti-X concentrator (Clontech), and then used to infect intestinal organoids.

**Mice**. Transgenic mice expressing the FRET biosensor for ERK (EKAREV mice) have been described previously[30]. FRET mice were backcrossed to C57BL/6N Jcl (CLEA Japan) for more than ten generations. To date, no disease or anomaly has been observed in these mice. Fucci2a mice were provided by the RIKEN BRC through the National Bio-Resource Project of the MEXT, Japan[45]. Apc$^{\Delta 716}$ mice have been reported previously[42]. Mice were housed in a specific pathogen-free facility and provided with a standard diet and water ad libitum. In some experiments, CHIR99021 (20 mg kg$^{-1}$ body weight, Cayman Chemical), erlotinib (100 mg kg$^{-1}$ body weight, Wako Pure Chemical Industries), and/or vehicle (dimethyl sulfoxide, DMSO) were intraperitoneally injected daily to 7-week-old C57BL/6 mice for 3 days. No statistical method was used to predetermine sample size. The experiments were not randomized. The investigators were not blinded to allocation during experiments and outcome assessment. The animal protocols were reviewed and approved by the Animal Care and Use Committee of Kyoto University Graduate School of Medicine (No. 10584).

**Microscopy**. For two-photon excitation microscopy (2PM), we used an FV1200MPE-IX83 inverted microscope (Olympus) equipped with a 30×/1.05 NA silicon oil-immersion objective lens (UPLSAPO 30XS; Olympus), an LCV110-MPE incubator microscope (Olympus) equipped with a 25×/1.05 water-immersion objective lens (XLPLN 25XWMP2; Olympus), and an InSight DeepSee Laser (Spectra Physics). The laser power was set to 3–18%. The scan speed was set between 4–12.5 μs per pixel. Z-stack images were acquired at 1–10 μm intervals. In time-lapse analyses, images were recorded every 1–3 min. The excitation wavelength for CFP was 840 nm. We used an IR-cut filter (BA685RIF-3), two dichroic mirrors (DM505 and DM570), and two emission filters (BA460-500 for CFP and BA520-560 for YFP) (Olympus). Confocal images were acquired with an FV1000/IX83 confocal microscope (Olympus) equipped with a 30×/1.05 NA silicon oil-immersion objective lens (UPLSAPO 30XS; Olympus).

**In vivo observation of the small intestine**. Intravital imaging of the small intestine was performed as described previously[66]. Mice were anaesthetized with 1.5–2% isoflurane (Abbott) inhalation and placed in the supine position on an electric heating pad maintained at 37 °C. Before surgery, the abdominal area of the mouse was disinfected using 70% ethanol. A small vertical incision was made on the right side of the abdominal wall. The small intestine was pulled out of the abdominal cavity, and both proximal and distal sides of the small intestine of interest were ligated using 5-0 surgical silk sutures (Nesco Suture). After PBS was administered into the intestinal cavity with a 29-gauge needle, the small intestine was put on a cover glass placed on a heat-stage maintained at 37 °C, and fixed with surgical sutures to minimize peristalsis. No drugs were used to stop peristalsis of the intestinal tract. Instead, we dilated the small intestine by injecting PBS into the cavity in order to minimize the peristalsis. Time-lapse imaging was then performed with an FV1200MPE-IX83 two-photon excitation microscope under control of the FluoView software (Olympus). In some experiments, 0.1 mg kg$^{-1}$ body weight of TPA (LC laboratories) or 5 mg kg$^{-1}$ body weight of PD0325901 (EMD Millipore) was administered via the caudal vein, and 100 mg kg$^{-1}$ body weight of erlotinib (Wako Pure Chemical Industries) was injected intraperitoneally during imaging. Mice were euthanized after the experiments.

**Intestinal organoid culture**. Small intestinal organoids were generated according to the published protocol[29]. Briefly, intestinal crypts were isolated from the mouse small intestine by incubation for 30 min at 4 °C in PBS containing 2 mM EDTA, and cultured in Advanced DMEM/F12 Medium (Thermo Fisher Scientific) supplemented with EGF (50 ng ml$^{-1}$, PeproTech), Noggin (100 ng ml$^{-1}$, PeproTech), and R-spondin1 (500 ng ml$^{-1}$, R&D Systems). Adenoma cells were isolated from the Apc$^{\Delta 716}$ mouse small intestinal adenomas by incubation for 60 min at 4 °C in PBS containing 2 mM EDTA. Organoids were embedded in Growth Factor Reduced Matrigel (Corning), and maintained by serial passaging. Organoids were imaged using an incubator-type two-photon excitation microscope, LCV110-MPE, within 4 days of the last passage in order to minimize the autofluorescence emitted from the cell debris accumulating in their central cavities. In some experiments, organoids were treated with TPA (1 μM, LC laboratories), PD0325901 (200 nM, Calbiochem), PD153035 (1 μM, MedChemexpress), CP-724714 (10 μM, AdooQ), Marimastat (100 μM, R&D Systems), or SB590885 (100 nM–10 μM, LKT Laboratories). For EGF starvation, organoids were cultured in media without EGF for 24 h. For pharmacological activation of Wnt signalling, organoids were cultured in media supplemented with CHIR99021 (5 μM, Cayman Chemical) for 24 h. For Wnt ligand stimulation, organoids were cultured in media supplemented with mouse recombinant Wnt3a (100 ng ml$^{-1}$, R&D Systems). We generated organoids expressing the dominant negative form of either EGFR or ErbB2 and adenomaderived organoids expressing EKAREV using the aforementioned lentiviruses as described previously[67]. All organoids were generated from mice housed in a specific pathogen-free facility, and assumed to be free from mycoplasma contamination.

**Image processing**. Acquired images were analyzed with MetaMorph software (Universal Imaging) as described previously[30,68]. Briefly, FRET efficiency was visualized as FRET/CFP ratio images shown in the intensity-modulated display mode, in which eight colours from red to blue represent the FRET/CFP ratio and the 32 grades of colour intensity represent the fluorescence intensity of CFP according to the colour scale shown in each figure. In order to analyze the Z-stack images, we performed maximum intensity projection, in which voxels with the maximum intensity at each $x$–$y$ coordinate were projected in the visualization plane. To reduce the noise of in vivo imaging data, a $5 \times 5$ median filter was applied to the FRET/CFP ratio images. In addition, time course data of the FRET/CFP values in each cell were smoothened by 6-min moving averages. Quantitative parameters of pulsatile ERK activation were obtained as follows. First, FRET/CFP values in each cell were calculated for all time frames. Second, the FRET/CFP values were smoothened by 6-min moving averages. Subsequently, the smoothed time series for each cell was fitted to either a flat line or a multi-peak function:

$$y(t) = \begin{cases} \omega_0 & (N = 0) \\ \max\{y_1(t), \ldots, y_N(t)\} + \omega_0 & (N \geq 1) \end{cases},$$

where $N$ and $w_o$ indicate the number of ERK activity pulses and basal ERK activity, respectively, max() is a function returning the maximum value of input variables and $y_i(t)$ indicates time-course of the $i$th ERK activity pulse represented by

$$y_i(t) = A_i \varphi_i(t - t_i),$$

where $A_i$ and $t_i$ are the amplitude and timing of the $i$th ERK pulse, respectively, and $\varphi_i(s)$ is a radial basis function. We here adopted a cosine function within a single period, i.e., $\varphi_i(s) = \cos(\omega_i s) + 1$ if $-\pi/\omega_i \leq s \leq \pi/\omega_i$ or $\varphi_i(s) = 0$ otherwise. Parameters ($A_i$, $t_i$, and $w_i$) were optimized by minimizing the square error between a fitting function and the observed ERK activity. If one of the amplitudes $A_i$ was greater than threshold, cells whose data were fitted to the multi-peak function were defined as "ERK-pulse$^+$", otherwise they were defined as "ERK-pulse$^-$". The data analysis was performed with MATLAB software (MathWorks).

**Immunofluorescence staining**. Small intestine specimens were washed with cold PBS, and fixed overnight in 10% formalin in PBS at 4 °C. The tissues were sequentially treated with PBS containing 12, 15, and 18% sucrose (for more than 2 h for each treatment), embedded in O.C.T. compound (Tissue-Teck), frozen, and sectioned at 8-μm thickness. Sections were subjected to immunofluorescence staining with the following primary antibodies: anti-EGFR (Medical & Biological Laboratories, clone 6F1), anti-ErbB2 (Cell Signaling Technology, clone 29D8), antiphospho-ErbB2 (Cell Signaling Technology, clone 6B12), anti-Ki67 (Abcam, ab15580), and anti-E-cadherin (Cell Signaling Technology, clone 24E10). The sections were treated with 0.5% Triton X-100 in TBS for 10 min. Antigen retrieval was performed by treating samples for 20 min in Tris-EDTA buffer, pH 8.0. Alexa Fluor 488 conjugated goat anti-rabbit IgG (Molecular Probes, no. A-11008), Alexa Fluor 546 conjugated goat anti-rabbit IgG (Molecular Probes, no. A-11035), or Alexa Fluor 488 conjugated goat anti-mouse IgG (Molecular Probes, no. A-11029; dilution: 1:500) were used as secondary antibodies. Counterstaining was performed with Hoechst 33342 (Molecular Probes, no. H3570). For EdU staining, mice were injected intraperitoneally with EdU (80 μg g$^{-1}$ body weight) 2 h before euthanization. Organoids were treated with 10 μM of EdU for 2 h before staining. EdU staining was performed using the Click-iT EdU Alexa Fluor 488 Imaging Kit (Life Technologies, no. C10337), according to the manufacturer's instructions.

**Extraction of RNA and qRT-PCR**. IECs were isolated by incubating and shaking intestinal specimens in cold PBS containing 5 mM EDTA. Total RNA was extracted from the isolated IECs or intestinal organoids by using an RNeasy Mini Kit (QIAGEN). The extracted total RNA was reverse-transcribed into cDNA by using a High-Capacity cDNA Reverse Transcription Kit (Thermo Fisher Scientific). Quantitative PCR analyses were performed by using an Applied Biosystems StepOne Real-Time PCR System and Power SYBR Green PCR Master Mix (Thermo Fisher Scientific). The expression data were normalized to those of GAPDH or β-actin.

**Microarray analysis**. For microarray analysis, we performed two independent series of experiments. Total RNA was extracted from intestinal organoids cultured in the ENR media supplemented with or without CHIR99021 (5 μM) for 4 days as described above. Synthesis of cDNA, in vitro transcription and labelling of cRNA, and hybridization to the Mouse Gene 1.0 ST arrays (Affymetrix, Santa Clara, CA, USA) were performed according to the manufacturer's protocols. The CEL files were analyzed by Transcriptome Analysis Console (TAC) 4.0 software (Thermo Fisher Scientific). Expression signals of all genes (probe sets) were calculated using the RMA algorithm. Gene set enrichment analysis was performed by using GSEA 3.0 software (Broad Institute)[69].

**Statistical analysis**. Mann–Whitney $U$-test was performed to analyze the statistical difference between two sets of data. Welch's $t$ test was performed when the number of samples was less than 6. As a multiple comparison test, the Steel–Dwass test was used after validation of homoscedasticity by the Bartlett's test. BH procedure was used for adjustment for multiple comparisons in $\chi^2$ test. $P$ values less than 0.05 were considered statistically significant. No statistical method was used to predetermine sample size. All statistical analyses were performed using the JMP Pro 13 software and R software (ver. 3.4.1).

**Data availability**. The data that support the findings of this study are available within the article and its Supplementary Information or from the corresponding author upon reasonable request. The microarray data have been deposited in the Gene Expression Omnibus (GEO) database under the accession code GSE110257 (https://www.ncbi.nlm.nih.gov/geo/query/acc.cgi?acc = GSE110257).

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

## Acknowledgements

We are grateful to H. Miyoshi (Division of Experimental Therapeutics, Graduate School of Medicine, Kyoto University) for technical advice. We thank K. Hirano, K. Takakura, S. Kobayashi, S. Fujiwara, N. Nishimoto, Y. Takeshita, A. Kawagishi, and the Medical Research Support Center of Kyoto University for technical assistance. We are also grateful to the members of the Matsuda Laboratory for their helpful input. M.M. was supported by the Nakatani Foundation, CREST JPMJCR1654 and JSPS KAKENHI Grant Numbers 15H02397, 15H05949 "Resonance Bio", and 16H06280 "ABiS". M.I. was supported by the Takeda Science Foundation and JSPS KAKENHI Grant Numbers 16K21106, 18H05100, and 18K06929.

## Author contributions

Y.M., M.I., K.A., T.C., H.S. and M.M. designed the project. Y.M. and M.I. performed all the experiments. Y.F. developed an algorithm for image processing. K.S. and M.M.T. developed transgenic mice. A.S. performed microarray analyses. Y.M. analyzed the data. Y.M., M.I., and M.M. prepared the manuscript. M.I. and M.M. supervised the study.

## Additional information

**Competing interests:** The authors declare no competing interests.

