## [Peer Review File · Nature Communications]

Reviewers' comments:

Reviewer #1 (Remarks to the Author):

This study reports on the dynamics of ERK activity in mouse intestinal epithelial cells, both in vivo and in reconstituted organoid culture. There are some aspects of this study that are very nice, such as the in vivo imaging of ERK pulses that extend previous studies by this group of authors. This demonstrates that as is the case with other epithelial cells, ERK displays a stochastic pulsatile pattern of activation, most likely due to autocrine signaling coupled to a positive feedback loop through ERK. However, parts of this manuscript are difficult to understand due to changes in culture conditions in individual experiments that are not always indicated in the text or figure legends. More serious is that most of the studies are descriptive in nature with a lack of targeted experiments that would reveal the mechanisms underlying their observations. Thus, their central conclusions are not supported by the data.

Their basic observations can be summarized as:

1. Mouse intestinal epithelial cells show spontaneous ERK pulses that are dependent on autocrine signaling through the EGFR both in vivo and in vitro.
2. Reconstituted organoids cultured in extraordinarily high concentrations of EGF lose pulsatile signaling and become refractory to EGFR inhibition, but still retain sensitivity to erbB2 inhibitors.
3. Organoids reconstituted from APC mice or organoids treated with CHIR99021 and high concentrations of EGF regain some degree of sensitivity to EGFR inhibitors.
4. Organoids reconstituted from APC mice or organoids treated with CHIR99021 appear to show altered EGFR distribution and expression levels as measured by immunofluorescence.
5. There appears to be a correlation between sensitivity to EGFR inhibitors and pulsatile ERK activity.

From this relatively limited set of observations, the authors make numerous claims and conclusions, especially on the role of WNT signaling in carcinogenesis and the role of different kinases in establishing ERK signaling patterns. Unfortunately, there is little concrete data to support these conjectures. The fundamental problem is that they do not understand the mechanistic basis of the ERK pulsatile signaling patterns and thus it is difficult, if not impossible, to understand what the different treatments are doing to perturb signaling in the cells. This is further complicated by the removal of EGF in some experiments (so that they can see the pulsatile patterns) and adding it back in others (to see "basal" ERK signaling). Furthermore, the authors appear to have not fully considered the numerous feedback loops that couple EGFR activation and ERK activation, which could provide alternative explanations to their data.

For example, it is known that there are both positive and negative feedback between ERK and the EGFR. ERK activation stimulates EGFR ligand shedding (e.g. TGF- α) in a positive feedback loop (doi: 10.1093/emboj/18.24.6962; doi: 10.1039/c003921g) while ERK can phosphorylate the EGFR, SOS1 and RAF in a negative feedback loop (e.g. doi: 10.1074/jbc.M110.148759; doi: 10.1038/nrm3048). Furthermore, ERK activation induces the synthesis of numerous negative regulators, such as the DUSPs, SPREDs, SPRYs, MIG6, which can shut off both EGFR and ERK signaling. At the same time, these negative regulators can have differential effects on EGFR versus erbB2 signaling. For example, MIG6 is specific for the EGFR. With multiple positive and negative feedback loops, there is a plethora of mechanisms occurring in their cells that could easily explain their observations, but are completely different than what they are proposing.

As an example of an alternate hypothesis, the chronic stimulation of cells with EGF could result in phosphorylation of the EGFR by ERK, resulting in a loss of EGFR-mediated signaling and the cells being refractory to EGFR inhibitors. However, kinase dead EGFR can still activate erbB2 through heterodimerization, thus explaining why the cells are still sensitive to erbB2 inhibition. The APC mutant or CHIR99021 treatment could induce the synthesis of a phosphatase (e.g. PTPN11), which

could remove the inhibitory phosphorylation of the EGFR, thus restoring the sensitivity of the cells to EGFR inhibitors. A dozen similar hypotheses could be generated, all consistent with their data, each with a totally different mechanistic basis. The point is that without any data that directly examines the feedback mechanisms occurring in their system and how they impact pulsatile signaling, it is not possible to say what is going on.

Finally, the authors confuse correlation with causation. Since EGF sensitivity is likely to be intrinsically coupled to pulsatile signaling through positive and negative feedback loops, you will always see a correlation between the two regardless of whether pulsatile signaling per se is needed for mitogenesis.

Minor points:

1. Need to be more specific on EGFR truncation. What residue?
2. EGFR and ERBB2 inhibitors are not totally specific. Although their data is consistent with a role of erbB2 in organoid physiology, they do not suggest how it is being activated. Is it through heterodimerization with the EGFR? If so, then they must assume that the EGFR is working as an inactive signaling partner. If through erbB3 heterodimerization, is there evidence for neuregulin expression?
3. The organoids are cultured in extremely high concentrations of EGF (50ng/ml), Noggin and R-spondin1. They show a pronounced effect of removing the EGF, but what about removing the other growth factors? How does this impact their results.
4. The use of the term "basal" activity is curious. They don't actually know how much pERK activity is present because they only do relative, population-based measurements. They need some direct biochemical measurements of pERK levels to calibrate their relative measurements.
5. Is the data from 4d in the presence or absence of EGF?
6. Are the experiments shown in Fig. 4e-4h in the presence or absence of EGF?

Reviewer #2 (Remarks to the Author):

Muta et al revealed the mechanism of ERK activation in intestinal epithelium using a combination of in vivo imaging and organoid culture. By intricate in vivo live imaging, they observed the existence of two modes of ERK activation in intestinal epithelium. Interestingly, ERBB2 signalling served to maintain basal ERK activation. Overall, the paper is well written and the conclusion is supported by both in vivo and in vitro evidence. I think the paper is interesting for broad readers. Nevertheless, I have some concerns regarding the last half part of paper as follows;

1. They showed that adenoma relied on EGFR signalling to greater extent than normal intestinal epithelium because of higher Wnt activation. This result was supported by CHIR treated organoids in the following experiments. However, some epigenetic changes (and possibly genetic changes other than Apc) might occur during the tumorigenesis, which might contribute to the favourable EGFR activation in adenoma. To determine whether the observed result was purely attributed to Apc mutation or was also implicated in other (epi)genetic changes concomitantly accrued during the tumorigenesis, the author should examine the same experiment using in vitro knockout Apc organoids (or isolated organoids immediately after in vivo Apc knockout before tumorigenesis).

2. In association with the above point, CHIR blocks GSK3beta and interferes various signalling other than Wnt/b-catenin signalling. The authors should check if the observed phenomenon also occurs with Wnt ligand stimulation.

3. The mechanism of the higher dependence on EGFR in adenoma was obscure. The detail analysis of molecular mechanism would be probably beyond the scope of this study. At least, the authors should show surface expression level of EGFR and ERBB2 on adenoma, CHIR-treated and non-treated normal organoids with or without EGF stimulation.

4. I am not sure if CHIR could really activate Wnt/b-catenin signalling in vivo. The authors should check their CHIR-treatment protocol can activate Wnt/b-catenin signalling in vivo. If not, the paper would be good enough without the CHIR in vivo treatment experiments.

Minor point.

1. line 348. They suddenly brought up HSP90, but what do you intend to say by this sentence?

Reviewer #3 (Remarks to the Author):

The manuscript by Muta et al is very well written and of a high standard necessary for Nature Communications. Here, the authors use elegant intravital ratio-metric FRET imaging to reveal pulsed versus basal signaling events in vivo and demonstrate the propagation of ERK signaling within intestinal crypt (in response to various clinically relevant drugs targeting EGFR/ErB2 etc) and in normal versus WNT altered conditions- increasing the fidelity and relevance of this work to maximise/understand limit of therapy and or how best to improve them. Moreover, the authors nicely complement this with use of crypt cultures in which distinct ligand/signal manipulations can pinpoint drivers of proliferation/response to therapy (due to ERK signaling dynamics) and also manage to show paradoxical feedback in certain conditions (ie braf). This work also reveals how we can monopolize upon altered ERK pulsing versus basal signaling in cancer cells (at the single cell level) that is not found in normal epithelial in the gut and this helps fine-tune therapy towards cancer versus normal, non-diseased contexts.

This manuscript is almost, in my opinion, ready to be published following a few key changes that need addressing.

1. Please in text both in the introduction and discussion address the known role of pulsed versus long term ERK signaling in PC12 cells to determine fate. There is a large body of work on this and this could also help link to the current phenomenon revealed here.
2. In the movies it is very clear there is propagation but to a non-expert reader it is hard to follow this..could the authors add tracking to outline the movement of this propagation in a side by side movie..ie one with non and other showing this movement, also possibly a kymograph could help here. The propagation in the crypt movies is great and this should be no problem to do- but will significantly help the wider audience/readership follow this MS
3. Fig 6 uses another biosensor application (Fucci) to compliment the proliferation effect/cell-cycle phase...did they measure crypt size? Or co-stain with anything in the experiment to back this up? Via IHC/IF- This would help solidify the phenomenon seen here. ie orthogonal standard assay would help this MS.
4. Similarly, could the authors for fig 7 do Ki67 or some basic orthogonal staining via IHC to confirm the result seen here...ie fig 7F is great I would like to see say any other simple survival/prolif staining to back this up.
5. Can the authors comment on whether any drugs were used to stop peristalsis for improved imaging? And if so, can they confirm this does not have off target effects re ERK signaling?
6. Please also acknowledge and caveat of this work and current use of EGFR in CRC...to provide a more balanced MS.
7. Can the authors also insert that this concept /phenomenon could be occurring for many other signaling nodes and targets and should be a future area of improved understanding in single cell drug response.

Point-by-point response:

Reviewer #1

This study reports on the dynamics of ERK activity in mouse intestinal epithelial cells, both in vivo and in reconstituted organoid culture. There are some aspects of this study that are very nice, such as the in vivo imaging of ERK pulses that extend previous studies by this group of authors. This demonstrates that as is the case with other epithelial cells, ERK displays a stochastic pulsatile pattern of activation, most likely due to autocrine signaling coupled to a positive feedback loop through ERK. However, parts of this manuscript are difficult to understand due to changes in culture conditions in individual experiments that are not always indicated in the text or figure legends. More serious is that most of the studies are descriptive in nature with a lack of targeted experiments that would reveal the mechanisms underlying their observations. Thus, their central conclusions are not supported by the data.

Their basic observations can be summarized as:

- 1. Mouse intestinal epithelial cells show spontaneous ERK pulses that are dependent on autocrine signaling through the EGFR both in vivo and in vitro.*
- 2. Reconstituted organoids cultured in extraordinarily high concentrations of EGF lose pulsatile signaling and become refractory to EGFR inhibition, but still retain sensitivity to erbB2 inhibitors.*
- 3. Organoids reconstituted from APC mice or organoids treated with CHIR99021 and high concentrations of EGF regain some degree of sensitivity to EGFR inhibitors.*
- 4. Organoids reconstituted from APC mice or organoids treated with CHIR99021 appear to show altered EGFR distribution and expression levels as measured by immunofluorescence.*
- 5. There appears to be a correlation between sensitivity to EGFR inhibitors and pulsatile ERK activity.*

From this relatively limited set of observations, the authors make numerous claims and conclusions, especially on the role of WNT signaling in carcinogenesis and the role of different kinases in establishing ERK signaling patterns. Unfortunately, there is little concrete data to support these conjectures. The fundamental problem is that they do not understand the mechanistic basis of the ERK pulsatile signaling patterns and thus it is difficult, if not impossible, to understand what the different treatments are doing to perturb

signaling in the cells. This is further complicated by the removal of EGF in some experiments (so that they can see the pulsatile patterns) and adding it back in others (to see “basal” ERK signaling). Furthermore, the authors appear to have not fully considered the numerous feedback loops that couple EGFR activation and ERK activation, which could provide alternative explanations to their data.

For example, it is known that there are both positive and negative feedback between ERK and the EGFR. ERK activation stimulates EGFR ligand shedding (e.g. TGF-alpha) in a positive feedback loop (doi: 10.1093/emboj/18.24.6962; doi: 10.1039/c003921g) while ERK can phosphorylate the EGFR, SOS1 and RAF in a negative feedback loop (e.g. doi: 10.1074/jbc.M110.148759; doi: 10.1038/nrm3048). Furthermore, ERK activation induces the synthesis of numerous negative regulators, such as the DUSPs, SPREDS, SPRYs, MIG6, which can shut off both EGFR and ERK signaling. At the same time, these negative regulators can have differential effects on EGFR versus erbB2 signaling. For example, MIG6 is specific for the EGFR. With multiple positive and negative feedback loops, there is a plethora of mechanisms occurring in their cells that could easily explain their observations, but are completely different than what they are proposing.

As an example of an alternate hypothesis, the chronic stimulation of cells with EGF could result in phosphorylation of the EGFR by ERK, resulting in a loss of EGFR-mediated signaling and the cells being refractory to EGFR inhibitors. However, kinase dead EGFR can still activate erbB2 through heterodimerization, thus explaining why the cells are still sensitive to erbB2 inhibition. The APC mutant or CHIR99021 treatment could induce the synthesis of a phosphatase (e.g. PTPN11), which could remove the inhibitory phosphorylation of the EGFR, thus restoring the sensitivity of the cells to EGFR inhibitors. A dozen similar hypotheses could be generated, all consistent with their data, each with a totally different mechanistic basis. The point is that without any data that directly examines the feedback mechanisms occurring in their system and how they impact pulsatile signaling, it is not possible to say what is going on.

Response:

First of all, we would like to express our gratitude to the reviewer for providing insightful comments and suggestions. As mentioned by the reviewer, EGFR/ERK signalling is regulated by complicated mechanisms comprising numerous regulators that often function in positive and negative feedback loops. Nevertheless, without considering involvement of these regulators, we raised an oversimplified model to explain the observation in the previous manuscript. We apologize for our insufficient analyses and explanation in the previous manuscript. In this revision, we have made our best efforts to address molecular mechanisms

underlying the observations, and identified several molecules that should be involved in the Wnt signalling-dependent regulation of EGFR/ERK signalling, as described below.

Since there are numerous possible candidates for EGFR/ERK signalling regulators that mediate altered ERK activity dynamics induced by Wnt signalling activation, we first performed unbiased, genome-wide screening of EGFR regulators whose expression levels are changed by Wnt signalling activation. To this end, we performed microarray analyses of control and GSK3 inhibitor (CHIR99021)-treated intestinal organoids. We identified 81 and 274 transcripts whose expression levels were significantly upregulated and downregulated, respectively, in CHIR99021-treated organoids by more than 3 folds (new Figure 8a). Notably, the following gene set enrichment analyses showed that gene expression changes induced by CHIR99021 resemble those observed in the *Apc* KO mouse intestinal epithelium and adenomas (new Figure 8b-e), suggesting that CHIR99021 treatment imitates Wnt signalling activation during intestinal tumorigenesis. We then inquired the expression levels of known EGFR/ERK signalling regulators, such as DUSPs, SPREDs, SPRYs, and MIG6, in the microarray data. These molecules are often involved in negative feedback regulation of EGFR/ERK signalling, and therefore important for the pulsatile nature of EGFR/ERK signalling, as suggested by the reviewer. The results showed that only *Dusp4* and *Dusp10* were upregulated by more than two folds (new Supplementary Figure 9a). However, these changes were not reproduced in the following RT-PCR analysis (new Figure 8f). We also investigated the expression levels of serine/threonine phosphatases, since they might antagonize ERK-dependent phosphorylation of EGFR, SOS1, and Raf. The phosphorylation of these proteins has been shown to act as negative-feedback regulation, as suggested by the reviewer. However, no serine/threonine phosphatase was included in the identified list of the CHIR99021-dependent upregulated and downregulated genes. Therefore, we concluded that CHIR99021 did not significantly affect expression of any of these regulators in our experimental settings. Thus, although these regulators might be involved in regulation of EGFR signalling, they are not likely to be a direct target of Wnt signalling in intestinal organoids.

We next searched our list of CHIR99021-dependent upregulated and downregulated genes for those genes whose protein products have been shown to regulate EGFR signalling. We found that the expression levels of four EGFR regulators (*Tnfrsf19* (Troy), *Flna*, *Egfl6*, and *Lrig3*) were significantly altered (new Supplementary table 1). RT-PCR analysis confirmed that expression of three factors (Troy, *Flna*, and *Egfl6*), which had been reported to promote EGFR signalling (Ding Z. et al., 2017, *Mol. Cancer Res.*; Fiori J.L. et al., 2009, *Endocrinology*; Chen J. et al., 2015, *Cancer Res.*), was upregulated, whereas that of *Lrig3*, which had been reported to suppress EGFR signalling (Guo D. et al., 2015, *J. Neurol. Sci.*), was downregulated by CHIR99021 treatment (new Figure 8f). Notably, expression of Troy, *Flna*, and *Egfl6* was upregulated, whereas that of *Lrig3* was downregulated, in adenoma organoids compared to normal organoids (new Figure 8g). Therefore, we hypothesized that the altered expression of these four factors might contribute to enhanced EGFR dependency of CHIR99021-treated organoids and adenoma organoids, and examined this idea by performing targeted knockdown or expression of these factors. We found that knockdown of

Egfl6 and expression of Lrig3 abolished a decrease in the basal ERK activity caused by EGFR inhibition in adenoma organoids (new Figure 8h). Moreover, knockdown of Troy and expression of Lrig3 decreased the frequency of ERK activity pulses in adenoma organoids (new Figure 8i). Knockdown of Flna did not significantly affect ERK activity dynamics (new Figure 8h, i). These results suggest that, in addition to elevated expression of EGFR (Figure 7), altered expression of multiple EGFR regulators, such as Troy, Egfl6, and Lrig3, coordinately enhances EGFR/ERK signalling in adenoma cells, thereby rendering these cells highly dependent on EGFR activity. Since most of conventional biochemical approaches cannot be applied to the intestinal organoid culture system due to difficulties in large scale culture and handling, elucidation of detailed molecular mechanisms by which the identified molecules enhance EGFR/ERK signalling possibly through affecting feedback mechanisms is beyond the scope of this study. We, rather, think that the most important conceptual advance in this study consists in elucidation (visualization) of ERK activity dynamics in the intestinal epithelium and their alteration during intestinal tumorigenesis. These findings and our approaches should provide an important insight for many researchers working on signal transduction research, as live-imaging is the only measure to track signalling activity dynamics at the single cell level. In addition to this, the reviewer's important comments led us to the identification of several regulators involved in Wnt-dependent regulation of EGFR/ERK signalling dynamics, which has greatly improved our study and should provide a clue about how ERK activity dynamics can be altered during tumorigenesis. We are grateful to the reviewer for raising these issues.

In addition to the above issues, we have also recognized that our previous manuscript contained insufficient expressions in several aspects. At first, changes in culture conditions (e.g. removal and addition of EGF) were not always described clearly, which made it difficult to understand our claims, as suggested by the reviewer. In the revised manuscript, we described the culture condition under which each experiment was performed. In addition, we also clearly mentioned that ERK activity pulses occurred spontaneously in individual cells, whereas propagation of these pulses to the adjacent cells was not observed, in the presence of EGF. We think that exogenous EGF supplemented in culture media decreases sensitivity of cells to the lower amount of EGFR ligands secreted by cells. Thus, propagation of ERK activity, which is likely mediated by shedding of endogenously expressed EGFR ligands, might be perturbed. In this process, negative feedback regulations of EGFR/ERK signalling might contribute to the decreased sensitivity of cells to EGFR ligands, as suggested by the reviewer. We mentioned these issues in the revised manuscript (page 8, line 16-22).

We also noticed that we need to clarify what exactly "EGFR signalling" means. As pointed out by the reviewer, EGFR can still activate ErbB2 through heterodimerization even when its kinase activity is inhibited. Without taking care of this function, we used a phrase "EGFR signalling" to express only signalling driven by the kinase activity of EGFR. In the revised manuscript, we explain this issue in the discussion section (page 18, line 3-8).

Finally, the authors confuse correlation with causation. Since EGF sensitivity is likely to be intrinsically coupled to pulsatile signaling through positive and negative feedback loops, you will always see a correlation between the two regardless of whether pulsatile signaling per se is needed for mitogenesis.

Response:

As the reviewer pointed out, our results do not demonstrate that the pulsatile nature of EGFR/ERK signalling per se is needed for mitogenesis. We do not intend to discuss the necessity of the pulsatile nature. Rather, we believe that pulsatile ERK activity, as well as the basal ERK activity, plays a role in promoting cell proliferation. Thus, we clarified this issue in the discussion section to avoid any misleading impression (page 16, line 29-page 17, line 3).

Minor points:

1. Need to be more specific on EGFR truncation. What residue?

Response:

The dominant negative form of EGFR is a C-terminally truncated mutant encoding amino acid 1 to 679, and the dominant negative form of ErbB2 encodes 1 to 693, respectively. We corrected the manuscript to clearly describe this in the method section (page 21, line 3-5).

2. EGFR and ERBB2 inhibitors are not totally specific. Although their data is consistent with a role of erbB2 in organoid physiology, they do not suggest how it is being activated. Is it through heterodimerization with the EGFR? If so, then they must assume that the EGFR is working as an inactive signaling partner. If through erbB3 heterodimerization, is there evidence for neuregulin expression?

Response:

We agreed with the reviewer's comments that EGFR and ErbB2 inhibitors are not necessarily specific. We thus performed shRNA-mediated knockdown of EGFR and ErbB2, and examined its effect on ERK activity dynamics. We found that knockdown of EGFR, but not that of ErbB2, decreased the frequency of ERK activity pulses (new Supplementary Figure

3f). In contrast, the basal ERK activity was decreased by the ErbB2 knockdown, but not by the EGFR knockdown (new Supplementary Figure 3b). These results corroborate the idea that kinase activity of ErbB2 mediates the basal ERK activity, whereas the pulsatile ERK activity depends on EGFR kinase activity. Since knockdown of EGFR did not significantly affect the basal ERK activity, which is dependent on ErbB2, we think that, in our experimental settings, ErbB2 does not necessarily require EGFR as a heterodimerization partner for its activation. In line with this, immunostaining of intestinal organoids with an antibody against phosphorylated ErbB2 (pErbB2) showed that an ErbB2 inhibitor decreased the amount of pErbB2, whereas an EGFR inhibitor did not affect it (new Supplementary Figure 8a-d). Thus, the majority of ErbB2 phosphorylation is likely to be mediated by ErbB2, but not by EGFR. We thus think that ErbB2 is activated mainly via homodimerization. However, we recognize that these data cannot completely rule out involvement of EGFR and ErbB3 in the ErbB2 activation process. We thus mentioned this possibility, as well as the above new data, in the revised manuscript (page 17, line 20- 26).

3. The organoids are cultured in extremely high concentrations of EGF (50ng/ml), Noggin and R-spondin1. They show a pronounced effect of removing the EGF, but what about removing the other growth factors? How does this impact their results.

Response:

From the viewpoint of the experts in growth factor signal transduction, the concentrations of these factors used in organoid culture may appear too high. In the present study, we followed the standard protocol developed by Dr. Hans Clevers lab. In accordance with the reviewer's suggestion, we examined the effects of Noggin or R-spondin1 removal on ERK activity dynamics and cell cycle progression in intestinal organoids. The results showed that ERK activity dynamics in organoids cultured in the Noggin-depleted (ER) or R-spondin1-depleted (EN) medium were comparable with those in the normal culture medium (ENR) (new Supplementary Figure 4a-f). The basal ERK activity was not affected by the removal of Noggin or R-spondin1 (new Supplementary Figure 4a, d). As observed under normal (ENR) culture condition, an ErbB2 inhibitor, but not an EGFR inhibitor, suppressed the basal ERK activity (new Supplementary Figure 4b, e), whereas the frequency of the ERK activity pulses was significantly decreased by an EGFR inhibitor, but not by an ErbB2 inhibitor, in organoids cultured in the ER or EN medium (new Supplementary Figure 4c, f). With regard to cell cycle progression, the proportion of cells in the S/G2/M phases was decreased by removal of Noggin and R-spondin1 (new Supplementary Figure 6a, b). These results indicate that, in intestinal organoids, R-spondin1 and Noggin are not required for ERK activation, but play an important role in promoting cell proliferation.

4. The use of the term “basal” activity is curious. They don’t actually know how much pERK activity is present because they only do relative, population-based measurements. They need some direct biochemical measurements of pERK levels to calibrate their relative measurements.

Response:

In accordance with the reviewer’s suggestion, we performed immunoblotting and live imaging in parallel under the same condition in order to calibrate FRET/CFP ratios to the fraction of phosphorylated ERK (pERK). The results showed that FRET/CFP ratios were well correlated with the pERK levels (new Supplementary Figure 2a), demonstrating that our ERK biosensor can monitor ERK activity in our experimental settings. Notably, under normal (ENR) culture condition, about 3% of total ERK was phosphorylated, which should mostly correspond to the basal ERK activity. The ERK activity was significantly decreased and increased by a MEK inhibitor and TPA, respectively (new Supplementary Figure 2a).

5. Is the data from 4d in the presence or absence of EGF?

6. Are the experiments shown in Fig. 4e-4h in the presence or absence of EGF?

Response:

Data shown in Fig. 4d and 4e-4h were obtained from organoids cultured in the presence of EGF (ENR medium). We explicitly described the culture condition in the revised manuscript and figures.

Reviewer #2

Muta et al revealed the mechanism of ERK activation in intestinal epithelium using a combination of in vivo imaging and organoid culture. By intricate in vivo live imaging, they observed the existence of two modes of ERK activation in intestinal epithelium. Interestingly, ERBB2 signalling served to maintain basal ERK activation. Overall, the paper is well written and the conclusion is supported by both in vivo and in vitro evidence. I think the paper is interesting for broad readers. Nevertheless, I have some concerns regarding the last half part of paper as follows;

1. They showed that adenoma relied on EGFR signalling to greater extent than normal intestinal epithelium because of higher Wnt activation. This result was supported by CHIR treated organoids in the following experiments. However, some epigenetic changes (and possibly genetic changes other than Apc) might occur during the tumorigenesis, which might contribute to the favourable EGFR activation in adenoma. To determine whether the observed result was purely attributed to Apc mutation or was also implicated in other (epi)genetic changes concomitantly accrued during the tumorigenesis, the author should examine the same experiment using in vitro knockout Apc organoids (or isolated organoids immediately after in vivo Apc knockout before tumorigenesis).

Response:

We recognize that adenoma cells might have some epigenetic and/or genetic changes other than the Apc mutation; therefore EGFR activation in those cells does not necessarily indicate that Wnt signalling activation can promote EGFR signalling. Since generation of Apc-knockout organoids is technically challenging due to low efficiency of viral infection and genome editing and also due to difficulties in cloning single intestinal stem cell, we performed expression of a constitutively active form of β -catenin (CA- β -catenin) to imitate Wnt signalling activation. The results have shown that overexpression of CA- β -catenin exerts effects similar to those of GSK3 inhibitor (CHIR) treatment on intestinal organoids: the basal ERK activity was suppressed by an EGFR inhibitor in the organoids expressing CA- β -catenin, but not in control organoids (new Supplementary Figure 5e). Moreover, overexpression of CA- β -catenin increased the frequency of ERK activity pulses, which was strongly suppressed by the EGFR inhibitor (new Supplementary Figure 5f). In addition to these results, we found that Wnt ligand stimulation also exerts similar effects on ERK activity dynamics and cell cycle progression (see the response to comment 2 described below). These results demonstrate that Wnt signalling activation alone can enhance EGFR signalling in intestinal epithelial cells. We are grateful to the reviewer for raising this issue.

2. In association with the above point, CHIR blocks GSK3beta and interferes various signalling other than Wnt/b-catenin signalling. The authors should check if the observed phenomenon also occurs with Wnt ligand stimulation.

Response:

In accordance with the reviewer's comment, we examined whether Wnt ligand stimulation could reproduce the effect of CHIR treatment. As expected, organoids stimulated with mouse recombinant Wnt-3a responded to EGFR and ErbB2 inhibitors similarly to CHIR-treated organoids: Basal ERK activity became more sensitive to EGFR inhibition (new Supplementary Figure 5b), and the frequency of ERK activity pulses was increased in an EGFR-dependent manner (new Supplementary Figure 5c). Furthermore, cell cycle analyses showed that Wnt-3a treatment increased the proportion of cells in the S/G2/M phases (new Supplementary Figure 6e). Taken together, our data showed that Wnt ligand stimulation, CHIR treatment, and overexpression of CA- β -catenin exert similar effects on ERK activity dynamics and cell cycle progression, demonstrating a role of canonical Wnt/ β -catenin signalling in these phenomena. We appreciate the reviewer's constructive suggestion.

3. The mechanism of the higher dependence on EGFR in adenoma was obscure. The detail analysis of molecular mechanism would be probably beyond the scope of this study. At least, the authors should show surface expression level of EGFR and ERBB2 on adenoma, CHIR-treated and non-treated normal organoids with or without EGF stimulation.

Response:

As pointed out by the reviewer, it should be important to examine the expression levels of EGFR and ErbB2 in intestinal organoids, for our previous data had suggested that increased expression of EGFR renders adenoma cells highly dependent on EGFR. To address this issue, we performed immunofluorescent staining of normal, CHIR-treated, and adenoma-derived organoids with anti-EGFR and anti-ErbB2 antibodies. Consistent with the in vivo distribution data (new Figure 7b), surface expression of EGFR was increased in CHIR-treated and adenoma-derived organoids compared to normal organoids, while that of ErbB2 was comparable among those three conditions (new Figure 7f, g, and Supplementary Figure 7c, d). We think that these results corroborate our idea that Wnt signalling activation enhances EGFR signalling partly through increasing EGFR expression. We are grateful to the reviewer for pointing it out.

4. I am not sure if CHIR could really activate Wnt/b-catenin signalling in vivo. The authors should check their CHIR-treatment protocol can activate Wnt/b-catenin signalling in vivo. If not, the paper would be good enough without the CHIR in vivo treatment experiments.

Response:

In order to evaluate the effect of CHIR on Wnt/ β -catenin signalling, we first performed gene expression profiling of control and CHIR-treated intestinal organoids by using microarrays (new Figure 8a). The results demonstrated that CHIR treatment induced gene expression changes similar to those occurred in APC-knockout and colorectal adenoma cells (new Figure 8b-e). As expected, CHIR-dependent upregulated genes included many previously reported Wnt-target genes. Among these genes, we focused on 5 most upregulated genes and inquired their expression levels in the intestinal epithelium of control and CHIR-treated mice. We found that most of those genes (4 out of 5 genes) were upregulated by CHIR (new Supplementary Figure 7a). In addition, the expression levels of a few established Wnt-target genes, such as EphB2 and EphB3, were also increased by CHIR (new Supplementary Figure 7a). Furthermore, immunofluorescent staining also revealed that expression of CD44, an established Wnt-target gene in the intestine, was enhanced in the crypt region of CHIR-treated mice (new Supplementary Figure 7b). Collectively, these results suggest that CHIR administration should activate Wnt/ β -catenin signalling in vivo.

Minor point.

1. line 348. They suddenly brought up HSP90, but what do you intend to say by this sentence?

Response:

We are sorry for our ambiguous expression in the previous manuscript. Although we intended to provide one of the possible explanations for the distinct regulation of EGFR and ErbB2 by citing HSP90 as an example of a molecule that might affect the stability and activity of ErbB2, this was, rather, confusing for readers. We have thus decided to delete the sentence from the manuscript.

Reviewer #3

The manuscript by Muta et al is very well written and of a high standard necessary for Nature Communications. Here, the authors use elegant intravital ratio-metric FRET imaging to reveal pulsed versus basal signaling events in vivo and demonstrate the propagation of ERK signaling within intestinal crypt (in response to various clinical relevant drugs targeting EGFR/ErB2 etc) and in normal versus WNT alter conditions- increasing the fidelity and relevance of this work to maximise/understand limit of therapy and or how best to improve them. Moreover, the authors nicely complement this with use of crypt cultures in which distinct ligand/signal manipulations can pin-point drivers of proliferation/response to therapy (due to ERK signaling dynamics) and also manage to show paradoxical feedback in certain conditions (ie braf). This work also reveals how we can monopolize upon alter ERK pulsing versus basal signaling in cancer cells (at the single cell level) that is not found in normal epithelial in the gut and this help fine-tuned therapy towards cancer versus normal, non diseased contexts.

This manuscript is almost, in my opinion, ready to be published following a few key changes that need addressed.

1. Please in text both in the introduction and discussion address the known role of pulsed versus long term ERk signaling in PC12 cells to determine fate. There is a large body of work on this and this could also help link to the current phenomenon revealed here.

Response:

We mentioned previous studies demonstrating importance of ERK signalling dynamics in cell fate decisions of PC12 cells as follows:

[Introduction, page: 4, line: 21-27]

Moreover, difference in ERK activity dynamics could also lead to different outputs in some biological process. An excellent example of such phenomena is neuronal differentiation of PC12 rat adrenal pheochromocytoma cells (Greene and Tischler, 1976, *PNAS*). Treatment of PC12 cells with NGF or FGF induces prolonged activation of ERK, which is essential for neuronal differentiation of these cells (Greene and Tischler, 1976, *PNAS*; Qiu and Green, 1992, *Neuron*). In contrast, treatment with EGF causes only transient, pulse-like ERK activation, and does not induce the differentiation (Qiu and Green, 1992, *Neuron*). Thus, difference in ERK activity dynamics resulted in different cell fate decisions in PC12 cells.

[Discussion, page: 18, line: 9-15]

Different dynamics of receptor tyrosine kinases could lead to distinct dynamics of the downstream ERK activity. For example, in PC12 cells, EGFR rapidly undergoes the classical clathrin-based endocytosis and the following degradation upon activation, whereas activation of TrkA, a receptor for NGF, induces its translocation to long-lived signalling endosomes, where TrkA avoids degradation and mediates sustained ERK activation (Valdez et al., 2007, *PNAS*). Thus, mechanisms regulating dynamics of specific receptor tyrosine kinases can be an important determinant of ERK activity dynamics.

We think that these changes would help wide-range of readers to recognize importance of understanding ERK activity dynamics and their possible mechanisms. We appreciate the reviewer's helpful suggestion

2. In the movies It is very clear there is propagation but to a non-expert reader it is hard to follow this..could the authors add tracking to outline the movement of this propagation in a side by side movies..ie one with non and other showing this movement, also possibly a kymograph could help here. The propagation in the crypt movies is great and this should be no problem to do- but will significantly help the wider audience/readership follow this MS

Response:

In accordance with the reviewer's suggestion, we added kymographs that show propagation of ERK activity pulses in the small intestinal epithelium and in intestinal organoids (new Supplementary Figure 1d, 2d). These changes have made our manuscript easier to follow for a broad readership. We are grateful to the reviewer for helpful suggestion.

3. Fig 6 uses another biosensor application (Fucci) to compliment the proliferation effect/cell-cycle phase...did they measure crypt size? Or co-stain with anything in the experiment to back this up? Via IHC/IF- This would help solidify the phenomenon seen here. Ie orthogonal standard assay would help this MS.

Response:

To address the issue raised by the reviewer, we performed EdU staining to quantify proliferating cells in the whole intestinal organoid. As expected, an ErbB2 inhibitor, but not an EGFR inhibitor, strongly decreased the number of EdU-positive cells (new Supplementary Figure 6d). Treatment with a GSK3 inhibitor, CHIR, slightly increased EdU-positive cells and this increase was cancelled by the EGFR inhibitor (new Supplementary Figure 6d). Combinatorial treatment with both EGFR and ErbB2 inhibitors most strongly decreased EdU-positive cells in both CHIR-treated and -non-treated organoids (new Supplementary Figure 6d). These results are well consistent with the data from our analysis with a Fucci cell cycle indicator (Figure 6h, i), and support distinct contribution of EGFR and ErbB2 to cell proliferation in intestinal organoids treated with or without CHIR. As for the size of intestinal organoids, we did not observe much difference between samples under our experimental condition, as we have focused on short-term effects of EGFR and ErbB2 inhibition in this study.

4. Similarly, could the authors for fig 7 do Ki67 or some basic orthogonal staining via IHC to confirm the result seen here...ie fig 7F is great I would like to see say any other simple survival/prolif staining to back this up.

Response:

In accordance with the reviewer's comment, we performed Ki67 staining to corroborate the data in Figure 7f (new Supplementary Figure 7f, g). The results demonstrated that administration of a GSK3 inhibitor, CHIR, promoted proliferation of crypt cells, and that an EGFR inhibitor suppressed cell proliferation only in mice administered with CHIR. These results strengthen our conclusion that Wnt signalling activation renders intestinal epithelial cells highly dependent on EGFR signalling. We are grateful to the reviewer for pointing it out.

5. Can the authors comment of whether any drugs were used to stop peristalsis for improved imaging? And if so, can they confirm this does not have off target effects re ERk signaling?

Response:

We did not use any drugs (except an anesthetic agent, isoflurane) to stop peristalsis of the intestinal tract. Instead, we dilated the small intestine by injecting PBS into the cavity in

order to minimize the peristalsis. In the revised manuscript, we clearly described it in the methods section (page 22, line 26-28).

6. Please also acknowledge and caveat of this work and current use of EGFR in CRC...to provide a more balanced MS.

Response:

We recognized that it is important to describe limitations of this work and current use of EGFR inhibitors in CRC treatment. At first, there are significant differences between adenomas developed in the mouse small intestine and human CRCs. In particular, the mouse adenomas are usually assumed to have only *Apc* mutations, whereas human CRCs often contain hundreds of genetic mutations, which might affect cellular responses to EGFR inhibition. Indeed, it has been already established that EGFR inhibitors are not effective against cancers harboring *RAS* or *RAF* mutations. In addition, CRCs, which are initially sensitive to EGFR inhibitors, can also acquire resistance to these drugs via several mechanisms. We described these caveats in the revised manuscript (page19, line 17-23).

7. Can the authors also insert that this concept /phenomenon could be occurring for many other signaling nodes and targets and should be a future area of improved understanding in single cell drug response.

Response:

In accordance with the reviewer's suggestion, we mentioned that similar phenomena could be occurred for many other signalling pathways and molecules, elucidation of which should be an important future research area to improve understanding of cellular drug responses at the single cell level (page 20, line23-26).

Reviewers' comments:

Reviewer #1 (Remarks to the Author):

This is a much improved manuscript describing the impact of altered Wnt signaling on the sensitivity of cells to EGFR inhibitors. The primary additions were documenting the impact of the Wnt pathway inhibitors on gene expression that could provide an underlying molecular explanation to the EGFR pathway crosstalk. This addition of mechanistic data substantially enhances the value of their primary observations and provides support for their hypotheses. They have also clarified aspects of the manuscript that should make the information less ambiguous and more accessible to other scientists. Overall, their findings on the crosstalk between the Wnt and EGFR pathway should be of broad interest.

However, the additions to the manuscript have created a few, relatively minor problems. Instead of revising the entire manuscript to include the new data and findings, the new material has simply been appended to the old manuscript. This has resulted in sometimes contradictory statements and an extremely long discussion section (from <4 pages to >5). For example, they state that "these results do not indicate that the pulsatile nature of EGFR-ERK signalling per se is needed for the promotion of cell proliferation. Our data only indicate that pulsatile ERK activity, as well as basal ERK activity, plays a role in promoting cell proliferation." I fail to see how something that is not needed can be stated to play a role. Indeed, they have not shown that the pulsatile nature of the ERK signaling plays any significant role in cell proliferation. They have only shown that there is a correlation between regaining activation of the EGFR, pulsatile signaling and cell proliferation. This is a critical point because the EGFR activates numerous downstream signaling pathways in addition to ERK signaling. The correlation itself could be explained by an increase in the total level of ERK signaling rather than the pulsatile pattern itself. For example, a very recent paper showed that the transcriptional control machinery downstream of ERK cannot discriminate between pulses and simply elevated levels of pERK activity (see Gillies et al. (2017). Linear Integration of ERK Activity Predominates over Persistence Detection in Fra-1 Regulation. *Cell Systems* 5, 549-563.e5). I have no objection to the authors stating that an increased frequency of ERK pulses is correlated with increased proliferation or that it could be mechanistically involved, but stating in the abstract that "the frequency of ERK activity pulses was also increased to promote cell proliferation" cannot be justified by the data they present.

As a corollary to the point about the new data simply being appended to the previous manuscript, the abstract has not been changed to reflect the new data. The abstract also need some editing. The statement "augmented EGFR signalling and exalted it to a dominant driver of ERK activity dynamics, which rendered IECs addicted to EGFR signaling" is both poor grammar and hyperbole. The phrase "deregulated activation of Wnt signaling" is needlessly vague. What they showed was that activation of Wnt signaling augmented EGFR signaling. I suggest that the authors rewrite the abstract to more accurately describe their actual findings rather than being a scientific editorial.

Reviewer #2 (Remarks to the Author):

My concern about CHIR-treatment in vivo phenotype remained. qPCR analysis lacked more faithful Wnt target gene, Axin2, and stem cell marker, Lgr5. Supplementary Figure 7b looks non-specific. If small intestinal epithelium properly responded to Wnt-activator, the intestine should show hyperplastic phenotype as reported by doi:10.1038/nature22313. Many researchers conceive CHIR treatment, including me, yet nobody convincingly demonstrated robust Wnt activation in small intestines.

Reviewer#3

Editorial note: Reviewer#3 expresses his satisfaction with the revised version in his confidential comments to the editor.

Point-by-point response:

Reviewer #1

This is a much improved manuscript describing the impact of altered Wnt signaling on the sensitivity of cells to EGFR inhibitors. The primary additions were documenting the impact of the Wnt pathway inhibitors on gene expression that could provide an underlying molecular explanation to the EGFR pathway crosstalk. This addition of mechanistic data substantially enhances the value of their primary observations and provides support for their hypotheses. They have also clarified aspects of the manuscript that should make the information less ambiguous and more accessible to other scientists. Overall, their findings on the crosstalk between the Wnt and EGFR pathway should be of broad interest.

However, the additions to the manuscript have created a few, relatively minor problems. Instead of revising the entire manuscript to include the new data and findings, the new material has simply been appended to the old manuscript. This has resulted in sometimes contradictory statements and an extremely long discussion section (from <4 pages to >5). For example, they state that "these results do not indicate that the pulsatile nature of EGFR-ERK signalling per se is needed for the promotion of cell proliferation. Our data only indicate that pulsatile ERK activity, as well as basal ERK activity, plays a role in promoting cell proliferation." I fail to see how something that is not needed can be stated to play a role. Indeed, they have not shown that the pulsatile nature of the ERK signaling plays any significant role in cell proliferation. They have only shown that there is a correlation between regaining activation of the EGFR, pulsatile signaling and cell proliferation. This is a critical point because the EGFR activates numerous downstream signaling pathways in addition to ERK signaling. The correlation itself could be explained by an increase in the total level of ERK signaling rather than the pulsatile pattern itself. For example, a very recent paper showed that the transcriptional control machinery downstream of ERK cannot discriminate between pulses and simply elevated levels of pERK activity (see Gillies et al. (2017). Linear Integration of ERK Activity Predominates over Persistence Detection in Fra-1 Regulation. Cell Systems 5, 549-563.e5). I have no objection to the authors stating that an increased frequency of ERK pulses is correlated with increased proliferation or that it could be mechanistically involved, but stating in the abstract that "the frequency of ERK activity pulses was also increased to promote cell proliferation" cannot be justified by the data they present.

As a corollary to the point about the new data simply being appended to the previous manuscript, the abstract has not been changed to reflect the new data. The abstract also need some editing. The statement "augmented EGFR signalling and exalted it to a dominant driver

of ERK activity dynamics, which rendered IECs addicted to EGFR signaling" is both poor grammar and hyperbole. The phrase "deregulated activation of Wnt signaling" is needlessly vague. What they showed was that activation of Wnt signaling augmented EGFR signaling. I suggest that the authors rewrite the abstract to more accurately describe their actual findings rather than being a scientific editorial.

Response:

We are grateful to the reviewer for the positive appreciation of our first revision. We agreed with the reviewer that our data have not shown the necessity of the pulsatile nature of EGFR signaling in cell proliferation but have only shown correlation between pulsatile EGFR signaling and cell proliferation. We thus corrected corresponding expressions in this second revision (page 3, line 11-12; page 16, line 25-page 17, line 6). Also, in accordance with the reviewer's comments, we carefully rewrote the abstract to reflect our new data and more accurately describe our findings. In addition, we have shortened the entire manuscript (including the discussion section) by deleting repetitive and nonessential sentences to keep the word limit of the journal.

Reviewer #2

My concern about CHIR-treatment in vivo phenotype remained. qPCR analysis lacked more faithful Wnt target gene, Axin2, and stem cell marker, Lgr5. Supplementary Figure 7b looks non-specific.

If small intestinal epithelium properly responded to Wnt-activator, the intestine should show hyperplastic phenotype as reported by doi:10.1038/nature22313.

Many researchers conceive CHIR treatment, including me, yet nobody convincingly demonstrated robust Wnt activation in small intestines.

Response:

We understood the reviewer's concern about the data on the effects of *in vivo* administration of CHIR. In accordance with the reviewer's suggestion, we compared the expression levels of Axin2 and Lgr5 in the intestinal epithelium of normal and CHIR-treated mice. The expression levels of Axin2 and Lgr5 were slightly elevated in CHIR-treated mice, however the differences were not statistically significant (Supplementary Figure 7a). Although Axin2 and Lgr5 are biologically important targets of Wnt signaling, their induction by CHIR is not very strong (2.47 and 4.35-fold in our microarray data, respectively) and therefore might be masked by a large variance in mouse experiments. With regard to the tissue phenotype, CHIR administration promoted cell proliferation only weakly (Figure 7h, i), and did not induce hyperplasia in the small intestine, as pointed out by the reviewer. Based on these results, we are thinking that the effect of *in vivo* CHIR administration is not as strong as genetic activation of Wnt signaling used in the previous studies.

We believe that our data convincingly showed that administration of CHIR exerts similar effects, at least, on ERK activity dynamics and expression of several genes both *in vitro* and *in vivo*. Moreover, our microarray data showed that, in intestinal organoids, CHIR treatment causes gene expression changes similar to those caused by Wnt signaling activation *in vivo*. Taken together, these data support the notion that the effect of CHIR treatment *in vivo* reflects some aspects of the Wnt signaling-mediated augmentation of EGFR signaling. At the same time, however, we also recognized that the effects of CHIR treatment are not completely the same as those of Wnt signaling activation, as many events other than GSK3 β inhibition occur in Wnt signaling and GSK3 plays various roles independently of Wnt signaling. Given all the circumstances described above, we thought that several expressions used in the previous manuscript might not be suitable to describe the results of this experiment. Since our data only showed that *in vivo* administration of CHIR induces changes in the ERK activity dynamics and expression of several genes, which are similar to those occurred *in vitro* (in intestinal organoids), the use of more general expressions, such as "pharmacological activation of Wnt signalling", is not appropriate and might be considered overinterpretation. We thus changed such expressions into phrases like "*in vivo*

administration of a GSK3 inhibitor” to express our findings more accurately and to avoid any misleading impression (page 13, line 7 and line 11; page 14, line 8; page 19, line 12). We also mentioned the above issues concerning differences between CHIR administration and Wnt signaling activation in the revised manuscript (page 19, line 14-21). We think that these changes have made our manuscript more appropriate and convincing. We are grateful to the reviewer for raising this issue. As for staining of CD44 (previous Supplementary Figure 7b), CD44 expression appeared to be enhanced in CHIR-treated samples compared to control samples. However, we recognized that the staining was weak and might not be convincing for other researchers. Since quantitative data from RT-PCR analysis should be more useful to see the effects of CHIR administration *in vivo*, we have decided to delete the CD44 staining data from the manuscript.

REVIEWERS' COMMENTS:

Reviewer #2 (Remarks to the Author):

My concerns have been addressed.